# Revisiting the high tropospheric ozone levels over Southern Africa: Roles of biomass burning and anthropogenic emissions

Yufen Wang[1], Ke Li[1*], Xi Chen[1], Zhenjiang Yang[1], Minglong Tang[1], Pascoal M.D. Campos[2], Yang Yang[1], Xu Yue[1], and Hong Liao[1]

[1]Jiangsu Key Laboratory of Atmospheric Environment Monitoring and Pollution Control, Jiangsu Collaborative Innovation Centre of Atmospheric Environment and Equipment Technology, Joint International Research Laboratory of Climate and Environment Change, State Key Laboratory of Climate System Prediction and Risk Management, School of Environmental Science and Engineering, Nanjing University of Information Science and Technology, Nanjing, China
[2]CNIC-Centro Nacional de Investiga ção Cient fica, Minist ério Do Ensino Superior Ci ência, Tecnologia e Inova ção, Avenida Ho Chi Minh N °201, Luanda, Angola

*Correspondence to: Ke Li (keli@nuist.edu.cn)

**Abstract.** Tropospheric ozone over Southern Africa is particularly high and causes tremendous health risks and crop yield losses. It has been previously attributed to the influence by biomass burning (BB), with a neglected contribution from anthropogenic emissions. However, due to the lack of measurements for ozone and its precursors, the modeled impacts of BB and anthropogenic emissions on tropospheric ozone levels in Southern Africa were not well evaluated. In this study, we combined the nested GEOS-Chem simulation with a horizontal resolution of $0.5 ° \times 0.625 °$ with available multiple observations at the surface and from space to quantify tropospheric ozone and its main drivers in Southern Africa. Firstly, BB emissions from current different inventories exhibit similar peaks in summer season but also have large uncertainties in Southern Africa (e.g., uncertainty of a factor of 2-3 in emitted $NO_x$). The model-satellite comparison in fire season (July-August) in 2019 shows that using the widely-used Global Fire Emissions Database Version 4.1 (GFED4.1) inventory, the model tends to overestimate by 87% compared to OMI $NO_2$, while the Quick Fire Emissions Database (QFED2) inventory can greatly reduce this model bias to only 34%. Consequently, the modeled tropospheric column ozone (TCO) bias was reduced from 14% by GFED4.1 to 2.3% by QFED2. In addition, the QFED2 also has a much better spatial representativeness than GFED4.1. The simulated surface daily maximum 8-hour mean (MDA8) ozone was decreased from 74 ppb by GFED4.1 to only 56 ppb by QFED2. This suggests a highly overestimated role of BB emissions in surface ozone if GFED4.1 is adopted. The model-observation comparison at the surface shows that the global Community Emissions Data System (CEDSv2) anthropogenic inventory tends to underestimate anthropogenic $NO_x$ emissions in typical Southern African cities and likely misrepresented anthropogenic sources in some areas. That means that urban ozone and $PM_{2.5}$ concentrations in Southern Africa may be strongly underestimated. For example, a ten-fold increase in anthropogenic $NO_x$ emissions can change ozone chemistry regime and increase $PM_{2.5}$ by up to 50 $\mu g\ m^{-3}$ at the Luanda city. Furthermore, we also find that the TROPOMI can already capture the urban $NO_2$ column hotspots over low-emission regions like Southern

Africa while this is unavailable from the OMI instrument, highlighting the critical role of high-resolution measurements in understanding atmospheric chemistry issues over Southern Africa. Our study presents a deeper understanding of the key emission sources and their impacts over Southern Africa that will be helpful not only to formulate targeted pollution controls, but also to enhance the capability in predicting future air quality and climate change, which would be beneficial for achieving a healthy, climate-friendly, and resilient development in Africa.

**Keywords:** biomass burning, tropospheric ozone, Southern Africa, urban emissions

## 1 Introduction

Tropospheric ozone ($O_3$) is an important trace gas in the atmosphere, posing multifaceted threatens to public health, crop yield, and global environment (Xu et al., 2018; Bourgeois et al., 2021). Complex photochemical reactions of nitrogen oxides ($NO_x = NO + NO_2$) and volatile organic compounds (VOCs) in the presence of sunlight is the main source of tropospheric ozone (Wang et al., 2022). These two ozone precursors are emitted from both anthropogenic and natural sources. Great efforts have been made to reduce anthropogenic emissions, but ozone pollution is still challenging in many urban regions across the globe (Gaudel et al., 2020; Lyu et al., 2023). Globally, it was estimated that ~365,000 premature deaths could be attributed to ozone pollution in 2019 (Murray et al., 2020). The urban population exposed to ozone was increased at a trend of 0.8% per year from 2000 to 2019, and the largest increases of daily maximum 8-hour mean (MDA8) ozone occurred in Africa and India (Sicard et al., 2023). However, due to the lack of comprehensive studies on tropospheric ozone pollution in Southern Africa, it is urgent to explore the major source contributions driving ozone pollution over these less-studied regions. A better understanding of the major emission sources is not only helpful to formulate actionable targeted pollution controls and to reduce air pollution risks, but also important to predict future air quality in developing regions under the rapid changing of emissions and climate change.

Biomass burning (BB) emits large amounts of air pollutants that are important ozone precursors (Qin et al., 2024). Africa is frequently exposed to intense BB (Vernooij et al., 2021), contributing to 70% of the global BB area and nearly 75% of global infant deaths attributed to BB pollutants (Jiang et al., 2020; Hickman et al., 2021). Exposure to air pollution from BB has strong differences in socioeconomic levels (Yue et al., 2024), with the most heavily exposed populations being in Southern Africa (Xu et al., 2023). Due to the complex climate types and unique lifestyles, BB in Africa during June-August months is concentrated over Southern Africa (Meyer-Arnek et al., 2005; Williams et al., 2010), and this "slash-and-burn" agricultural activity could lead to the very high ozone concentrations over Southern Africa. As shown in **Figure S1**, surface ozone concentrations in Southern Africa were simulated exceeding 100 ppb in July, making it to be the highest ozone level worldwide. This is consistent with the previous modeling findings that BB activities are the dominant driver of tropospheric

ozone in this region (V. Clarmann et al., 2007). At the city level, Rwanda with observed daily ozone maximum of 70 ppb during the dry season (Dewitt et al., 2019) can be affected by the transport of BB from Northern and Southern Africa. The high ozone is mainly driven by BB $NO_x$ emissions; for example, the Southern African BB can increase $NO_x$ concentrations by a factor of 2-5 in July-August months (Hoelzemann, 2006). Although BB has a great impact on ozone and its precursors in Southern Africa, there are few quantitative studies on this issue.

The popular way of quantifying the role of BB is to conduct chemical transport modeling, e.g., using the ECHAM5-MOZ (Aghedo et al., 2007), GEOS-Chem (Wang et al., 2022; Marvin et al., 2021), and WRF-Chem (Yang et al., 2022). The ECHAM5-MOZ model simulations show that BB can increase surface ozone by more than 50 ppb in Central Africa in June-August during 1997-2001 (Aghedo et al., 2007). Williams et al. (2010) used the Tracer Model version 4 to simulate June-August air pollution in 2006, and they showed that BB in Southern Africa is the largest source of carbon monoxide and ozone precursor emissions in Africa. However, model assessment is highly dependent on BB emission inventories and there is a lack of comparative studies of different BB inventories over the Southern Africa. This is because existing BB emission inventories have large uncertainties in Africa (Petrenko et al., 2012; Shi et al., 2015). The most widely-used inventory for global model simulations is the Global Fire Emissions Database (GFED) (Shi et al., 2020), and other BB inventories include the Quick Fire Emissions Database (QFED), the Global Fire Assimilation System (GFAS), and the Fire Inventory from NCAR (FINN). The uncertainties of a factor of 2-10 among these inventories source from estimated burned area, emission factors, and vegetation type (Fu et al., 2022). Depending on how fire emissions are calculated, these inventories can be divided into two categories: the fuel-based bottom-up estimation (e.g., GFED and FINN) (Pechony et al., 2013; Nikonovas et al., 2017) and the satellite-derived top-down estimation (e.g., QFED and GFAS) (Nikonovas et al., 2017). In addition, the injected height of BB emissions is also a key factor in determining the residence time of pollutants in the atmosphere that would impact the spatiotemporal distribution of tropospheric ozone (Rémy et al., 2017). Therefore, it is urgent to take advantage of observational constraints to evaluate the current BB inventories and quantify their impacts on tropospheric ozone in Africa.

In addition to the effects of BB, tropospheric ozone can be also affected by anthropogenic emissions in Africa. Although the intensity of anthropogenic emissions is relatively low in Africa, its impact at the urban scale cannot be ignored. With the rapid urbanization (Liousse et al., 2014), mean concentrations of surface $SO_2$, $PM_{2.5}$, and $PM_{10}$ in Luanda have exceeded European Union human health protection limits (Campos et al., 2021). More importantly, anthropogenic emissions (e.g., black carbon) are projected to be comparable with BB emissions by 2030 in Africa (Liousse et al., 2014). Projection studies also pointed out that 50% of the population will be expected to live in cities by 2050 (Aucoin and Bello-Schünemann, 2016), resulting in a significant increase in the population exposure to ozone in Africa. With air pollution becoming a major

cause of premature deaths in Africa (Julien et al., 2018), urban air pollution will likely pose more challenges in the context of increasing anthropogenic emissions (Roy, 2016; Marais and Wiedinmyer, 2016; Zhang et al., 2021). However, whether the current anthropogenic emission inventories are reasonable in urban Southern Africa remains unclear.

Lack of surface observations is a major challenge for assessment of emission inventories in Africa. About 66% of African countries do not have regular air quality monitoring (Fajersztajn et al., 2014), particularly few in Southern Africa (Julien et al., 2018). Recently, there are continuous surface measurements available at several major cities over Southern Africa (**Figure 1**), together with the high-resolution satellite observations (e.g., TROPOMI), which could be very helpful to detect urban air pollution in this region. With the worsening air pollution in Africa (Sicard et al., 2023), it is particularly timely to take advantage of these valuable measurements to assess the key drivers of high tropospheric ozone in Southern Africa.

As abovementioned, there are notable uncertainties in the estimation of major emission sources over Southern Africa. Assessing and predicting the impacts of emissions on air quality and health risks rely heavily on model simulations, and these uncertainties in emission inventories can affect the development of effective control strategies. We therefore need to utilize surface and satellite observations to gain a comprehensive understanding of the emission source contributions in Southern Africa. This will help to develop effective mitigation measures to realize the Sustainable Development Goals for having a healthy, climate-friendly, and resilient development in Africa.

Here we integrated the high-resolution GEOS-Chem model and newly-available measurements to estimate the impact of biomass burning and anthropogenic emissions on tropospheric ozone over Southern Africa. The aim of this study is: 1) to quantify the role of BB emissions on regional tropospheric ozone over Southern Africa, and 2) to assess the representativeness of anthropogenic emission inventories in urban Southern Africa and their impacts on urban ozone pollution. Observational data, model description and experimental setup are presented in Section 2. Section 3.1 shows the major emission sources and the high tropospheric ozone issue over Southern Africa. The estimated impacts of biomass burning and anthropogenic emissions on tropospheric ozone are analysed in Sections 3.2 and 3.3, respectively. Conclusions and discussion are given in Section 4.

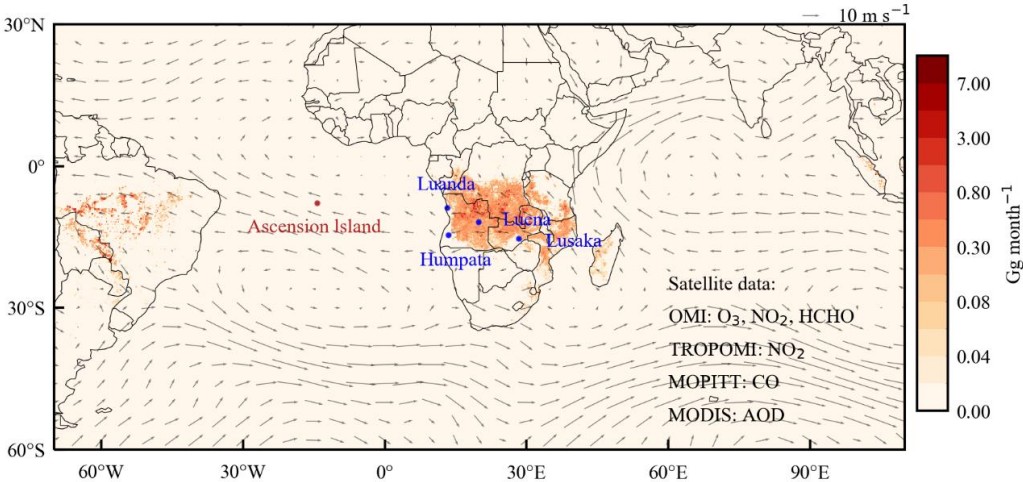

**Figure 1.** The biomass burning $NO_x$ emissions (shaded) and 850hpa wind fields. The BB $NO_x$ emissions are for July-August 2019 from the GFED4.1 inventory (unit: Gg month$^{-1}$). Blue dots represent the locations of surface observations and red dot denotes the ozonesonde measurement; and satellite data used in this study are listed in the lower right corner.

## 2 Measurement data and model description

### 2.1 Surface measurements

**Figure 1** shows the locations of the four surface observation sites over Southern Africa. Hourly and daily real-time air quality indexes (AQI) for $NO_2$ and $PM_{2.5}$ were obtained from the Worldwide Air Quality Index (https://aqicn.org/station). The AQI can be converted to pollutant concentrations based on the website's AQI Calculator. These four sites record continuous measurement data in the study area, namely: Humpata in Angola (14 °95' S, 13 °44' E), Luanda in Angola (8 °80' S, 13 °23' E), Luena in Angola (11 °76' S, 19 °91' E), and Lusaka in Zambia (15 °41' S, 28 °29' E). The stations in Angola and Zambia have been operating since mid-May 2023 and February 2022, respectively, and data for June-August 2023 were selected for this study. To evaluate modeled ozone profiles, we adopted the Ascension Island's ozonesonde data from the Southern Hemisphere Additional Ozone Sounding (SHADOWZ) network, which measured ozone profile from 1998 (Thompson et al., 2000).

### 2.2 Satellite data

In order to investigate the model results driven by different BB emission inventories and the anthropogenic emission inventories, multiple observations from the OMI (https://disc.gsfc.nasa.gov/datasets/), TROPOMI (https://www.earthdata.nasa.gov/sensors/tropomi), MODIS (https://ladsweb.modaps.eosdis.nasa.gov/search/), and MOPITT (https://giovanni.gsfc.nasa.gov/giovanni/) satellite instruments were used (Table 1). The OMI sensor observes

the globe once a day and is capable of obtaining the column concentration distributions of a variety of tropospheric trace gases (e.g., $NO_2$ and $O_3$). The TROPOMI sensor is a troposphere-specific observational instrument, and Wang et al. (2020) have compared the $NO_2$ concentrations of OMI and TROPOMI with observations. As for MODIS AOD, Shi et al. (2019) compared observations from 400 stations of the Aerosol Robotics Network (AERONET) with the MODIS AOD, and demonstrated that the MODIS was able to better capture the spatial and temporal variations of AERONET AOD (Zhang et al. 2024). MOPITT was launched in December 1999 on board the Earth observation satellite Terra with a 10:30 am equator crossing time (Kopacz et al., 2010). Here, as listed in **Table 1**, we used tropospheric ozone, $NO_2$, and HCHO observations from OMI with resolutions of $1°\times1.25°$, $0.25°\times0.25°$, and $0.05°\times0.05°$, respectively, as well as $NO_2$ observations from TROPOMI with a resolution of $0.125°\times0.125°$. AOD and CO observations with a resolution of $1°\times1°$ are from MODIS and MOPITT, respectively. We sampled the model simulation results consistent with satellite overpass times in the following comparisons. Although these satellite datasets have been well employed to reflect emission changes, their uncertainties are also notable due to biases in slant column density, air mass factor, and stratosphere-troposphere separation. For example, the reported uncertainties in $NO_2$ columns from OMI and TROPOMI are 25-50% and they can be increased to 50-100% in terms of OMI HCHO columns.

**Table 1**. Satellite and surface observations used in this study.

| Species | | Spatial resolution/ site locations | Observation period |
|---|---|---|---|
| $O_3$ | OMI | $1°\times1.25°$ | July-August 2019 |
| $NO_2$ | OMI | $0.25°\times0.25°$ | 2019-2020 |
| HCHO | OMI | $0.05°\times0.05°$ | July-August 2019 |
| $NO_2$ | TROPOMI | $0.125°\times0.125°$ | 2018-2023 |
| AOD | MODIS | $1°\times1°$ | July-August 2019 |
| CO | MOPITT | $1°\times1°$ | July 2019 |
| PM$_{2.5}$ | Humpata | (14°34' S, 13°26' E) | June-August 2023 |
| | Luanda | (8°48' S, 13°14' E) | |
| | Luena | (11°45' S, 19°54' E) | |
| | Lusaka | (15°24' S, 28°17' E) | |
| $NO_2$ | Humpata | (14°34' S, 13°26' E) | June-August 2023 |
| | Luanda | (8°48' S, 13°14' E) | |
| | Luena | (11°45' S, 19°54' E) | |
| $O_3$ | Ascension Island | (7°58' S, 14°24' W) | July-August 2017-2019 |

## 2.3 Biomass burning emission inventories

In this study, six BB emission inventories were compared: GFED4.1, GFED5, GFAS, QFED2, FINNv1.5, and FINNv2.5. The GFED4.1 inventory provides dry matter emissions based on the area of BB and vegetation types from MODIS observations (Marvin et al., 2021; Zhang et al., 2018). The GFED5 is an updated version of GFED4.1 and the GFED5 global burned area is 61 % higher than GFED4.1 (Chen et al., 2023). The GFAS inventory estimates the amount of dry matter burning based on fire radiative power (FRP) (Vongruang et al., 2017). The QFED2 inventory is based on the FRP method and draws on the cloud correction method developed in GFAS with the high spatiotemporal resolution. FINNv1.5 calculates dry matter combustion using fire hot spots (FHS) data to calculate the burned area, and FINNv2.5 builds on this with extensive updates to the burned area, vegetation types, and chemicals emitted. In particular, FINNv2.5 adopted the active fire detections from the Visible Infrared Imaging Radiation Suite (VIIRS) to better capture small fires, and used multiple satellite products for daily fire emissions estimates (i.e., MODIS + VIIRS fire detections). The estimated BB $NO_x$ emissions from these inventories will be further discussed in Section 3.2.1.

## 2.4 GEOS-Chem Model

The atmospheric composition in Africa was simulated by using the nested version of the three-dimensional global chemical transport model (GEOS-Chem, version 13.3.3; http://acmg.seas.harvard.edu/geos/), which was driven by the Modern-Era Retrospective analysis for Research and Applications version 2 (MERRA-2) meteorological reanalysis dataset. The model domain was for Africa (35 °S - 30 °N, 17 °W - 50 °E) with a horizontal resolution of 0.5 ° × 0.625 ° and a vertical configuration of 47 layers. The chemical boundary conditions for the nested simulation are provided by the global GEOS-Chem simulation with a horizontal resolution of 2 ° × 2.5 °, which was updated every three hours. GEOS-Chem model includes fully coupled ozone-$NO_x$-hydrocarbon-aerosols chemistry mechanisms. $PM_{2.5}$ components include sulfate, nitrate, ammonium, dust, sea salt, organic carbon (OC), and black carbon (BC) (Park et al., 2004).

In Africa, anthropogenic emissions are from the Community Emissions Data System (CEDS) (Hoesly et al., 2018) and biogenic emissions are from the Model of Emissions of Gases and Aerosols from Nature (MEGAN) version 2.1 (Guenther et al., 2012). We simulated hourly concentrations of ozone, $NO_2$, and other pollutants in Africa using the nested GEOS-Chem model with a set of sensitivity simulations (**Table 2**). Here we focused our experiments on July-August 2019 for all sensitivity and benchmark simulations. In order to investigate the effect of BB on tropospheric ozone, we conducted three model experiments. Firstly, we used GFED4.1 and QFED2 inventories to simulate the hourly air pollutant concentrations for July-August, 2019 (Run_GFED and Run_QFED), respectively, and validated the model results with satellite observations. Then, we conducted a sensitivity experiment by scaling down the QFED $NO_x$ emissions to be consistent with

satellite $NO_2$ observation (Run_QFED_66%$NO_x$). We used two different BB inventories and turned off aerosol chemistry to explore the effect of aerosols on ozone (Run_GFED_no-aerosol and Run_QFED_no-aerosol). We also explored the effect of emission height on simulated tropospheric ozone by emitting BB pollutants only within the PBL (Run_QFED_PBL).

After evaluating the BB emission inventories at the regional scale, we set up a series of experiments to explore the impact of anthropogenic emissions on tropospheric ozone in Southern Africa. Considering that surface air pollutant measurements are only available for June-August 2023, and then we used the up-to-date QFED inventory to simulate concentrations of $NO_2$ and $PM_{2.5}$ in June-August 2023 (Run_QFED_2023) and compared them with five surface air quality observations. We also conducted model simulations for January-February 2020 (Run_QFED_2020) to explore the effect of anthropogenic emissions on tropospheric ozone during the non-fire season. It is noted that we fixed the anthropogenic emissions from CEDS at 2019 in all these simulations due to the lack of up-to-date anthropogenic emission data. Based on the underestimation of surface $NO_2$ observations in the model, we explored the sensitivity of ozone and $PM_{2.5}$ concentrations to anthropogenic $NO_x$ changes by a factor of 10 or 20 over the Southern Africa (Run_QFED_Anth10$NO_x$ and Run_QFED_Anth20$NO_x$); at the city scale, we explored the effects of perturbing anthropogenic $NO_x$ emissions in Luanda by a factor of ten for difference sectors (i.e., power plant, industrial, and transportation) (Run_QFED_Anth_10$NO_x$ _Sector).

Finally, we conducted two set of sensitivity simulations to attribute ozone to different emission sources, by turning off BB emissions, natural emissions (i.e., biogenic VOC and soil $NO_x$), and anthropogenic emissions, respectively (**Table 2**). In particular, we compared the ozone source attribution between the simulations by using the CEDS inventory and 10-fold CEDS $NO_x$ emission.

**Table 2**. GEOS-Chem model simulations.

| | Experiments | BB emissions | Anthropogenic emissions |
|---|---|---|---|
| Impacts of biomass burning (July-August 2017-2019) | Run_GFED | GFED4.1 | CEDSv2 |
| | Run_QFED | QFED2 | CEDSv2 |
| | Run_QFED_66%NO$_x$ | 34% reduction in QFED2 NO$_x$ emissions | |
| | Run_GFED_no-aerosol | Aerosol chemistry was turned off | |
| | Run_QFED_no-aerosol | | |
| | Run_QFED_PBL | 100% emissions below the PBL* | |
| Impacts of anthropogenic emissions | Run_QFED_2023 | | CEDSv2 |
| | Run_QFED_Anth10NO$_x$ | QFED2 June-August 2023 | 10-fold NO$_x$ emissions |
| | Run_QFED_Anth20NO$_x$ | | 20-fold NO$_x$ emissions |
| | Run_QFED_Anth_10NO$_x$_Sector | | 10-fold NO$_x$ emissions of energy, industry, and transportation sectors, respectively |
| | Run_QFED_2020 | QFED2 January-February 2020 | CEDSv2 |
| Ozone source attribution (July-August 2019) | Run_QFED_noBB | BB emissions were turned off | |
| | Run_QFED_noNatl | BVOC and soil NO$_x$ emissions were turned off | |
| | Run_QFED_noAnth | Anthropogenic emissions were turned off | |
| | Run_QFED_Anth10NO$_x$_noBB | BB emissions were turned off | |
| | Run_QFED_Anth10NO$_x$_noNatl | BVOC and soil NO$_x$ emissions were turned off | |
| | Run_QFED_Anth10NO$_x$_noAnth | Anthropogenic emissions were turned off | |

*The baseline simulation follows the vertical distribution of QFED2 emission (i.e., 65% emissions below the PBL and 35% emissions into the free atmosphere).

# 3. Results and discussion

## 3.1 Emission sources and simulated high ozone over Southern Africa

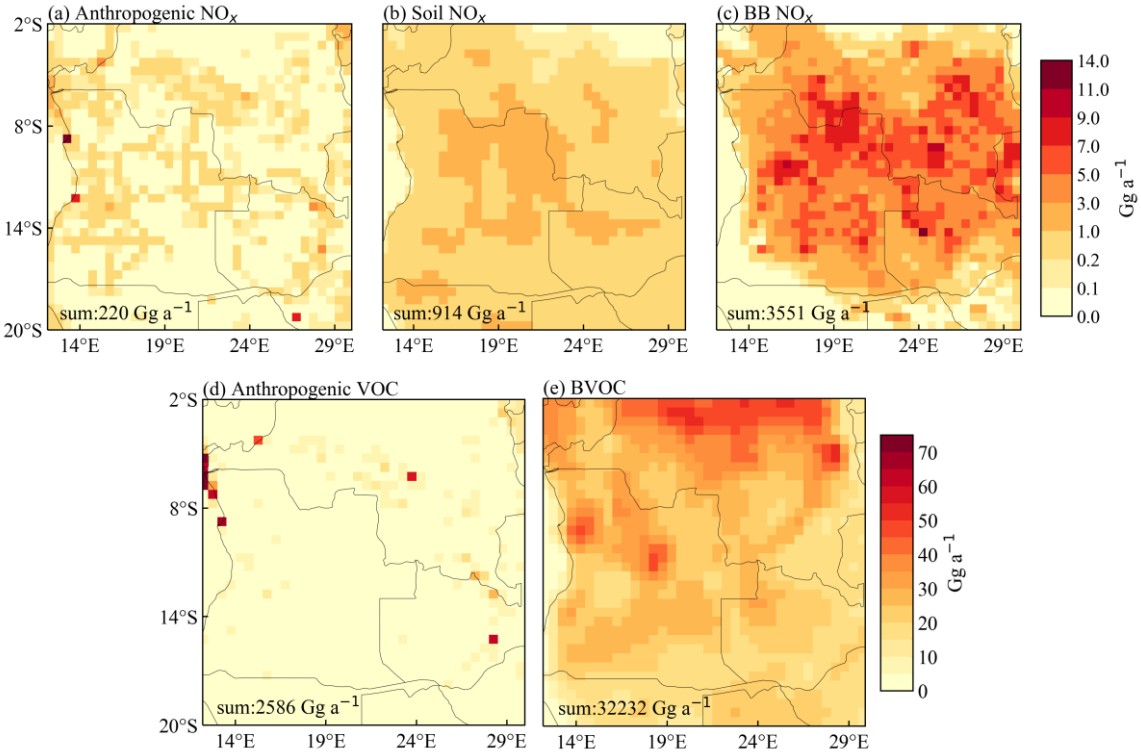

**Figure 2.** Spatial distributions of annual emissions of anthropogenic $NO_x$, soil $NO_x$, biomass burning $NO_x$, anthropogenic VOC, and BVOC in 2019 (unit: Gg $a^{-1}$). Anthropogenic $NO_x$ and VOC are from CEDSv2 inventory, soil $NO_x$ and BVOC are calculated by the GEOS-Chem model, and biomass burning $NO_x$ and VOC are from GFED4.1 inventory.

We compared the annual emissions of anthropogenic $NO_x$, soil $NO_x$, BB $NO_x$, as well as anthropogenic VOCs (AVOC) and biogenic VOCs (BVOC) over Southern Africa in 2019, as presented in **Figure 2**, which were estimated at 220 Gg $a^{-1}$, 914 Gg $a^{-1}$, 3551 Gg $a^{-1}$, 2586 Gg $a^{-1}$, and 32,232 Gg $a^{-1}$, respectively. It should be noted that here BB $NO_x$ emissions are from the GFED4.1 inventory. In terms of $NO_x$ emissions, BB emission is the largest contributor and is about 16 times of $NO_x$ emissions from anthropogenic sources. The regions with high anthropogenic emissions are mainly Luanda, Kinshasa, and Lusaka which are the capitals of Angola, the Democratic Republic of the Congo (DRC), and Zambia, respectively. High vegetation cover in Southern African region leads to high BVOC emissions which are about 12 times of AVOC emissions.

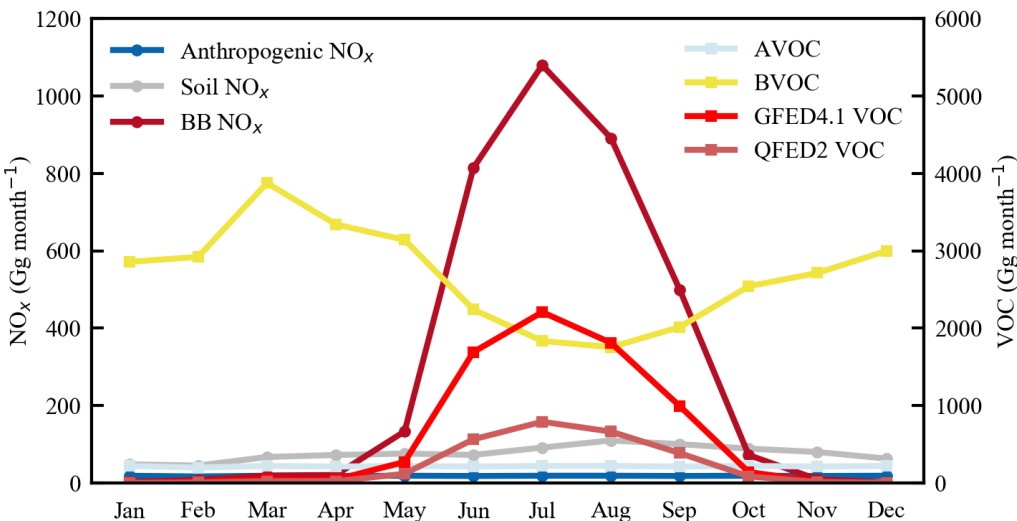

**Figure 3.** Seasonal variations in anthropogenic $NO_x$ (deep blue), soil $NO_x$ (grey), biomass burning $NO_x$ (red), biomass burning VOC (red), anthropogenic VOC (blue), and biogenic VOC (yellow) emissions in 2019 (unit: Gg month$^{-1}$). Anthropogenic $NO_x$ and VOC are from CEDSv2 inventory, soil $NO_x$ and BVOC are calculated by the GEOS-Chem model, and biomass burning $NO_x$ and VOC are from GFED4.1 and QFED2 inventory.

Seasonally, **Figure 3** presents the monthly variations of ozone precursor emissions averaged over Southern Africa in 2019. The BVOC emission exhibits a strong seasonal pattern ranging from 2000 Gg month$^{-1}$ to 4000 Gg month$^{-1}$, and it peaks in March and then decreases to the minimum in July-August. This seasonality is consistent with the seasonal variation in isoprene emissions in Southern Africa in 2006 as reported by Williams et al. (2009). We can also see that $NO_x$ from BB peaks during June-August, which is consistent with the results of Boschetti and Roy (2008). Emissions of BB $NO_x$ in January-April and November-December were relatively small. BB VOC has similar seasonal variability in both inventories, but the GFED4.1 inventory emits 2-3 times as much as the QFED2 inventory in fire season. The BVOC emissions are generally higher than BB VOC emissions except for those in July-August months from the GFED4.1 inventory. The seasonal contrast in BB $NO_x$ and BVOC emissions highlights the importance of BB in the production of high summer tropospheric ozone in this region (Vieira et al., 2023).

Simulated spatial distribution of MDA8 ozone in Africa from July to August 2019 obtained by using the GEOS-Chem model and the GFED4.1 inventory (Run_GFED) as in **Figure 4a**. The regional average of MDA8 ozone in Southern Africa is about 74 ppb and the maximum can be up to 120 ppb in northern Angola and southwest Congo. Dewitt et al. (2019) observed a daily ozone maximum of 70 ppb during the dry season in Rwanda, which is adjacent to the DRC, in 2015-2017.

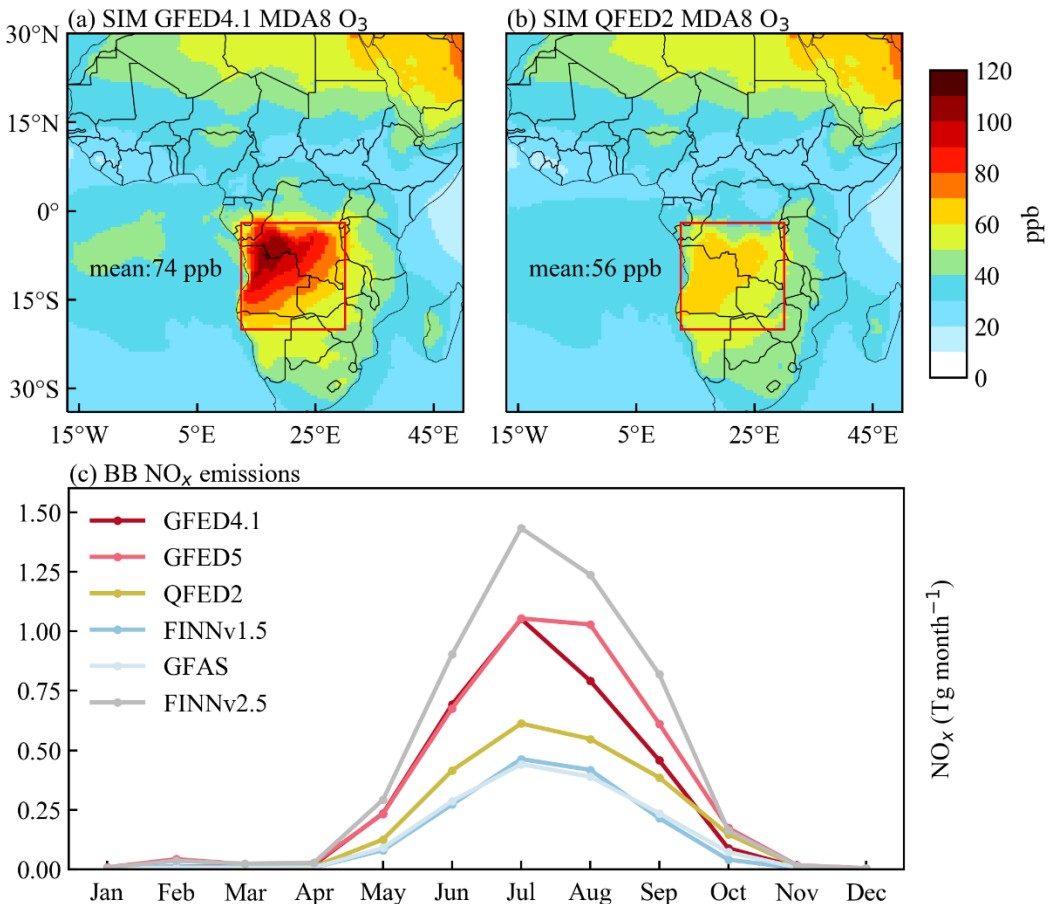

**Figure 4.** Differences in BB NO$_x$ emissions and modeled surface ozone from different inventories. (a-b) Surface MDA8 ozone simulated by GEOS-Chem model for July-August 2019 by the GFED4.1 and QFED2 inventories, respectively. (c) Monthly BB NO$_x$ emissions in 2014 averaged over the Southern African region.

Based on our simulation results, it can be found that the daily maximum ozone during the BB season is 86 ppb for Rwanda in the Run_GFED run, compared to only 62 ppb in Run_QFED run. Compared to the observed ozone in Rwanda, it may indicate an overestimation in the baseline simulation (Run_GFED). **Figure 5a** shows the spatial distribution of simulated tropospheric column ozone concentrations (TCO), with maximum values of up to 50 DU mainly in northern Angola and southwest Congo. Higher TCO levels are also seen over the Atlantic Ocean, which are mainly associated with long-range transport (Williams et al., 2010; Meyer-Arnek et al., 2005). Also, in Figure S2, we find that the GEOS-Chem simulated (Run_QFED) and OMI tropospheric ozone columns are in good agreement over the Atlantic Ocean after

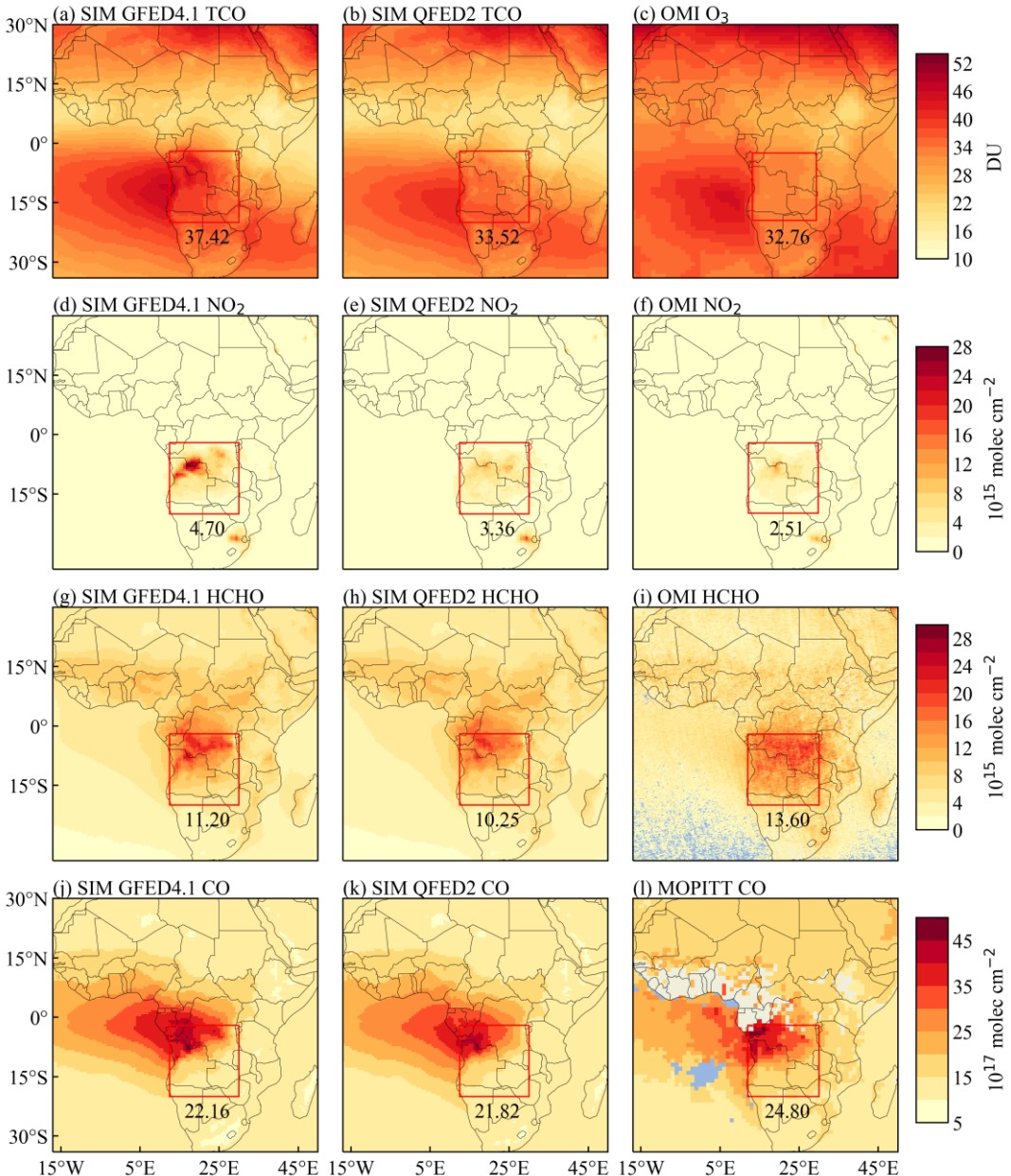

**Figure 5.** The comparison of GEOS-Chem simulated (left and middle panels) and satellite-based (right panels) tropospheric columns for ozone and its precursors. The simulated TCO, NO$_2$, HCHO, and CO columns in Africa for July-August 2019 were driven by the GFED4.1 and QFED2 inventories, respectively. For CO satellite data, only the July value was used due to the large amount of missing measurement in August. The numbers in the plots are the mean values in the red boxed area.

individually subtracting the background ozone values. Considering the strong seasonal variation of surface ozone in Southern Africa (Figure S3) and the estimated ozone precursors from different sources in Figure 3, here the large differences in simulated surface ozone with different BB inventories demonstrate that BB contributes greatly to high ozone concentrations during the fire season in Southern Africa. As such, a better understanding of the high ozone over Southern Africa would depend on the accurate estimate of BB emissions.

## 3.2 Impacts of biomass burning (BB) on tropospheric ozone

### 3.2.1 Uncertainties in BB emission inventories

Although the GEOS-Chem model has been widely employed for modeling tropospheric ozone globally (Balamurugan et al., 2021; Li et al., 2023), its evaluation against measurements over Southern Africa is very limited. In order to accurately evaluate the effects of BB emissions on tropospheric ozone, we need to take the uncertainties from different BB emissions into account (Wiedinmyer et al., 2023).

We compared the monthly emissions of BB $NO_x$ in 2014 for the Southern Africa from the six emission inventories (GFED4.1, GFED5, QFED2, GFAS, FINNv1.5, and FINNv2.5), as illustrates in **Figure 4c**. All of the six BB inventories share the similar seasonality in $NO_x$ emissions, but there are large differences with a factor of 2-3 in estimated emission intensities, particularly in the dry season. The inventory was divided into two groups based on the level of emissions, with the high emission groups being FINNv2.5, GFED5, and GFED4.1. FINNv2.5 shows the highest BB $NO_x$ emissions, which are 45% higher than GFED4 emissions and 130% higher than QFED2 emissions, but Wiedinmyer et al. (2023) also suggests that FINNv2.5 probably tends to overestimate $NO_x$ emissions in Africa. GFED5 is an updated version from GFED4.1, and their difference in $NO_x$ emissions is minimum in January-July and enlarged in August. The low emission groups are: QFED2, GFAS, and FINNv1.5. GFAS and FINNv1.5 resemble in the estimated $NO_x$ emissions but both of them are significantly lower than the other inventories. This lower estimate in the bottom-up FINNv1.5 inventory may be attributed to the underestimated burned area and emissions (Wiedinmyer et al., 2011) , and the lower top-down GFAS estimate could be due to the smaller emission factors (Liu et al., 2020). Therefore, in the following, we will use the GFED4.1 and QFED2 inventories to represent the high estimate and low estimate of BB $NO_x$ emissions for Southern Africa, respectively.

Spatially, there are also evident differences among different biomass burning inventories (Figure S4). The spatial distribution of the high values in GFED4.1 and QFED2 is generally consistent with a spatial correlation coefficient of 0.76, both showing high emissions in northeastern Angola. In contrast, the GFED5 inventory has high $NO_x$ emissions concentrated in southwestern Congo, and its spatial distribution differs considerably with QFED2. The GFAS inventory has a similar spatial distribution with QFED2 (a correlation coefficient of 0.84), but GFAS cannot capture the localized

high emissions as shown in QFED2 and GFED4.1. However, the FINNv1.5 and FINNv2.5 exhibit a very different spatial distribution compared to other inventories, with low emissions in Angola and high emissions in the Congo region. Their spatial correlation coefficients with the QFED2 inventory are 0.06 and 0.31, respectively.

In addition to $NO_x$ emissions, the VOC emissions are the highest in GFED5 and FINNv2.5 inventories, and the other four inventories show much smaller VOC emissions. Each inventory adopts different specific ratios for emitted chemical species, but they also differ with each other. For example, there is a $NO_x$/OC ratio of 1:0.6 in GFED4.1, 1:1.5 in GFED5, GFAS, and FINNv1.5, 1:3 in QFED2, and 1:1 in FINNv2.5 (Figure S5).

### 3.2.2 Simulated tropospheric ozone with different BB emissions

The simulated spatial distribution of MDA8 ozone in Africa during the fire season (July-August) in 2019 with the GFED4.1 and QFED2 inventories is given in **Figure 4a** and **Figure 4b**, respectively. The simulated surface MDA8 ozone by the GFED4.1 inventory is 74 ppb over Southern Africa, which is 32% higher than the value of 56 ppb by the QFED2 inventory. The maximum value of MDA8 ozone by the GFED4.1 inventory can reach up to 120 ppb, but the maximum value by the QFED2 inventory is only 70 ppb. This remarkable discrepancy suggests that the uncertainties in BB emissions could play an important role in simulating surface ozone over Southern Africa. This is consistent with previous findings that BB emissions lead to strong ozone increases in Southern Africa during the fire season (V. Clarmann et al., 2007; Jaffe and Wigder, 2012). For the tropospheric ozone, **Figures 5a-5b** show the simulated spatial distribution of TCO by using the GFED4.1 and QFED2 inventories. In contract to surface ozone, the regional average of TCO simulated by the GFED4.1 inventory is only 4 DU (11%) higher than that simulated by the QFED2 inventory. We will show the simulated ozone difference between these two inventories is mainly caused by BB $NO_x$ emissions, while BB VOC emissions only impact ozone levels slightly (Figure S6).

To evaluate the model performance in simulating the vertical profile of tropospheric ozone in Africa, we compared the model results with ozonesonde observations from Ascension Island, UK (7 96'S, 14 91'W) in **Figure 6**. As shown in **Figure 1**, Ascension Island is located downwind of the high BB area, and ozone and its precursors from BB can be transported from Southern Africa to the South Atlantic (Mari et al., 2008), leading to ozone enhancement in Ascension Island (Jenkins et al., 2021). The ozone concentrations modeled by GEOS-Chem respond well to the ozonesonde observations in terms of vertical distribution, and in particular the model captures the variation in observations with altitude well. The differences in ozone vertical distribution due to the two BB inventories are notable in the troposphere below 6 km, in particular at the altitude range of 3-6 km (Figure S7). Compared to the ozonesonde observations, this bias can be also found while GEOS-Chem captures the vertical ozone variations well regardless of which inventory is used. This is

consistent with the results of small TCO differences in **Figure 5**.

### 3.2.3 Satellite constraints on BB emission estimates

In order to evaluate the tropospheric ozone simulation in Africa region, in **Figure 5** and Table S1 we compared the simulated columns with GFED4.1 and QFED2 inventories against satellite observations of TCO and ozone precursors (e.g., $NO_2$, CO, and HCHO). The simulated TCO with GFED4.1 inventory shows high values of up to 50 DU near the fire source regions in northern Angola and southern DRC, and in the downwind region over Atlantic Ocean. The OMI TCO has the regional average of 37.4 DU, suggesting an overestimation of 14% in the GFED4.1 simulation relative to OMI. In contrast, the simulated TCO with QFED2 inventory is strongly spatially consistent with the OMI satellite, with a slight overestimation of 2.3%.

We also compared the simulated and observed tropospheric $NO_2$ columns in **Figures 5d-5e**. The GFED4.1 inventory simulation exhibits high value of up to $28 \times 10^{15}$ molecule $cm^{-2}$ near the BB source region, but there is a large overestimation of 87% with respect to the OMI satellite data. Similar conclusions are also from Anderson et al. (2021) that the model using the GFED4.1 inventory can capture high $NO_2$ in Africa but the bias was as high as 100%. This is in agreement with previous studies that model simulations trend to produce a high bias towards BB activities in Africa (Souri et al., 2024). However, the QFED2 inventory simulation can greatly reduce this high bias, with an overestimation of only 34%. In Figure S8, we also compared model results with the TROPOMI satellite, and a similar high bias was also found in the modeled $NO_2$ columns. If we further have QFED2 $NO_x$ emissions reduced by 34%, as shown in Figure S9, it can effectively reduce the bias for $NO_2$ columns from 34% to 0.4% and reduce the overestimation of the TCO columns to 1.1%. It is noted that this comparison between the simulated and satellite-based tropospheric columns could be biased due to their different representativeness in vertical profiles of chemical species. Anyway, this sensitivity simulation demonstrates the importance role of BB $NO_x$ emissions in tropospheric ozone production. In contrast, we find that the BB VOC emissions from GFED4.1 inventory are about 3 times the QFED2 inventory in fire season, but the regional mean changes are only 2.5 ppb for MDA8 ozone and 0.94 DU for TCO for July-August 2019 in response to a tripled QFED2 VOC emissions (Figure S6).

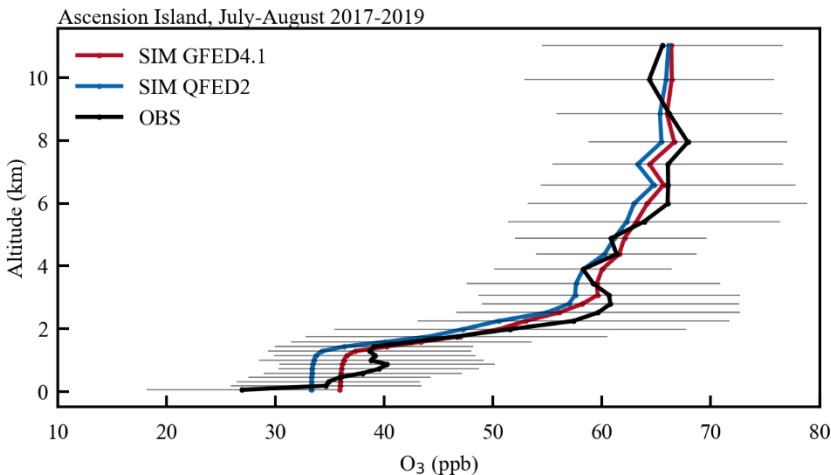

**Figure 6.** The comparison of GEOS-Chem simulated and measured vertical ozone distributions over the Ascension Island, UK, for July-August 2017-2019. The model results by the GFED4.1 (red) and QFED2 (blue) inventories are both given.

HCHO is one of the important VOCs in the troposphere, and a comparison of simulated and satellite-derived tropospheric HCHO columns is given in **Figures 5g-5i**. The HCHO column concentrations simulated by GEOS-Chem and the satellite observations exceeded $20 \times 10^{15}$ molecule cm$^{-2}$ in northern Angola and southwest DRC, and the underestimated HCHO columns in GEOS-Chem might be due to some missing VOC species (Zhao et al., 2024) and the lower anthropogenic NO$_x$ emissions in Southern Africa that both affect the chemical production of HCHO. Simulated HCHO column concentrations between the GFED4.1 and QFED2 inventories were consistent spatially, with only a difference of $1 \times 10^{15}$ molecule cm$^{-2}$ on a regional basis. The levels and spatial distributions of HCHO are mainly influenced by BVOC and BB emissions. Firstly, the Congo Basin, as one of the largest tropical rainforests, emits a large amount of BVOCs that can be oxidized to generate high values of HCHO (Wells et al., 2020). It leads to the spatial distribution of HCHO similar to the distribution of BVOC sources. Secondly, BB is found to be one of the main sources of HCHO in the African continent (Liu et al., 2020). Differences in VOC and NO$_x$ emissions between GFED4 and QFED inventories (Van Der Werf et al., 2017), e.g., BB VOC emissions in GFED4.1 being two times of the QFED2 inventory in 2019, may account for the slightly different HCHO columns.

The simulated CO columns in **Figures 5j-5l** are spatially similar to MOPPIT retrievals, with high values in the downwind regions of fire sources. The regional average of CO column concentrations simulated by GEOS-Chem is underestimated by approximately 10% compared to MOPITT, which reflects a long-lasting issue of CO underestimation in GEOS-Chem model (David et al., 2019; Ni et al., 2018). Hoelzemann (2006) used a variety of BB emission inventories to drive the

MOZART model to simulate CO concentrations in Southern Africa in September-October 2000, and they showed that all simulations exhibited an underestimation against the MOPITT CO. In addition, we also found that the simulated spatial distribution of CO columns is similar with each other among different BB inventories, and their regional difference is only 1%. This suggests that neither HCHO nor CO is the main reason for the overestimation of ozone production.

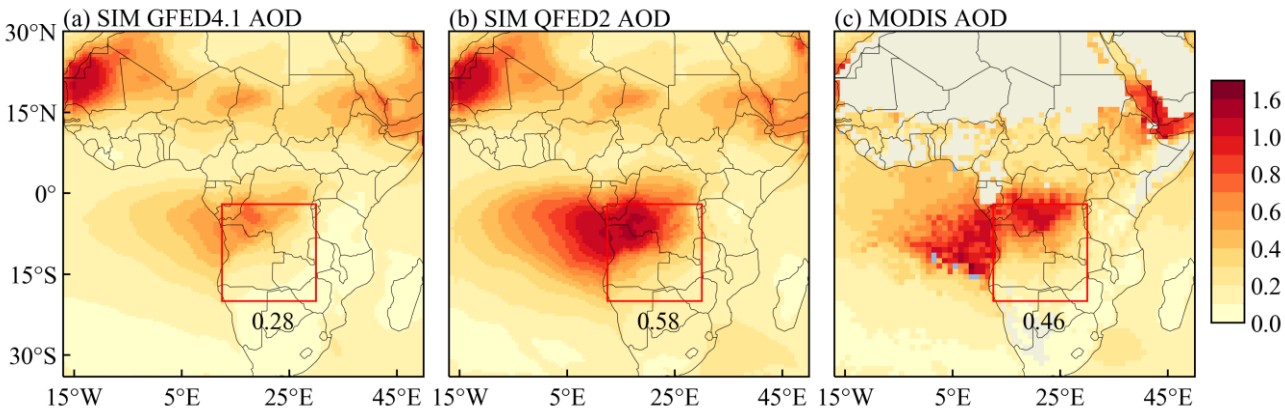

**Figure 7.** The comparison of GEOS-Chem simulated AOD in Africa in July-August 2019 with the MODIS AOD. The model results by the GFED4.1 (left) and QFED2 (middle) inventories are both given.

We compared the spatial distribution of modeled and satellite-based AOD as shown in **Figure 7**. The simulation results by both inventories can capture the spatial variability of MODSI AOD. But simulated regional mean AOD by the QFED2 inventory overestimated MODIS AOD by 26%, while the GFED4.1 inventory underestimated MODIS AOD by 37%. Tian et al. (2019) used GFED4 as an input to drive the GEOS-Chem model and also showed that the model tended to underestimate the intensity and spatial distribution of AOD in the African region. The inconsistency between these two inventories may be attributed to the discrepancy in carbonaceous aerosol emissions, since the OC and BC emissions from GFED4 are only half of the QFED emissions (Chang et al., 2023). In addition, the difference between OC and BC in the biomass burning emission inventories could affect ozone simulation through aerosol chemistry, and the results are shown in Figure S10. Aerosol chemistry mainly influences ozone formation by altering photolysis and heterogeneous processes. On the one hand, aerosol can change the shortwave radiation reaching the ground through scattering and absorption, which in turn affects the photolysis rate. On the other hand, aerosol can update reactive radicals (e.g., $HO_2$, nitrogen radicals) that are critical for ozone formation. After turning off aerosol chemistry alone in the model, regional surface ozone was increased by 10 ppb and TCO by 2 DU using the GFED4.1, while using the QFED2 regional ozone was increased by 14 ppb and TCO by 4 DU. As such, the lower level of aerosols in GFED4.1 may be a reason for the overestimation of simulated ozone concentrations.

In conclusion, the widely-used GFED4.1 inventory has a large bias in simulating tropospheric ozone in Southern Africa, and the QFED2 inventory exhibit much more consistent with satellite observations in terms of simulated concentration levels and spatial distributions (Table S1). This bias is mainly due to the overestimation of $NO_x$ emissions in Southern Africa in GFED4.1. Firstly, $NO_x$ emissions in GFED4.1 are 38% higher than in QFED2 in Southern Africa. Secondly, the modeled $NO_2$ column in GFED4.1 shows a high bias compared to QFED2 and satellite observations, while the modeled HCHO and CO columns are generally consistent between GFED4.1 and QFED2 inventories. Thus, we conclude that the overestimation of ozone in Southern Africa simulated with GFED4.1 is due mainly to the overestimation of $NO_x$ and the lower aerosol levels in GFED4.1 may be a minor reason for the overestimation of modeled ozone concentrations. We will use the QFED2 inventory for BB emissions in the following analysis.

### 3.2.4 Role of BB emission heights in ozone simulation

The representativeness of BB emission injection heights is also an important factor that can impact ozone simulations (Rémy et al., 2017). We conducted a sensitivity experiment using the QFED2 inventory and allowed all BB emissions emitting below the PBL. In **Figure 8**, the impact of this vertical partitioning on surface ozone varies regionally. At the surface, the changes of MDA8 ozone were within ±2.4 ppb and the BB source areas showed a decreased ozone. For TCO, the simulated mean values with this vertical partitioning were 0.2 DU higher than those without vertical partitioning, but the magnitude of this effect is smaller than the TCO changes (~4 DU) caused by the difference in BB $NO_x$ emissions between GFED4.1 and QFED2 inventories. Thus, our simulations demonstrate that the configuration of BB emission height has a limited effect on surface ozone level but a moderate influence on TCO columns in this region.

### 3.3 Impacts of anthropogenic emissions on tropospheric ozone

### 3.3.1 Uncertainties in anthropogenic emission inventories

Uncertainties may exist in anthropogenic emissions from regional scale to urban cities in Southern Africa. For example, in **Table 3** we compared the differences in $NO_x$ emissions between two widely-used global inventories: Community Emissions Data System (CEDSv2) and HTAPv3. Whether in Southern Africa or Luanda, there is a missing seasonality in $NO_x$ emissions in CEDSv2, whereas $NO_x$ emissions in HTAPv3 are much higher in January-February than in other months. Over the Southern Africa, monthly $NO_x$ emissions in the CEDSv2 are about 30% lower than the HTAPv3 in January-February. For Luanda, the CEDSv2 inventory is 87% lower than HTAPv3 in January-February 2018 and 20-50% lower in the other months. In addition, the validation of anthropogenic emission in global inventories was barely evaluated in this region and we will take advantage of recently available surface measurements and satellite retrievals to fill this gap.

### 3.3.2 Model evaluation against surface measurements of NO$_2$ and PM$_{2.5}$

Currently, there are very few surface observations in Southern Africa. However, in the study, there are three cities (Humpata, Luanda, and Luena) that have continuous surface measurements of NO$_2$ during the period of June-August 2023, and four cities (Humpata, Luanda, Luena, and Lusaka) with surface measurements of PM$_{2.5}$. These measurements are critical to understand the hotspots of urban anthropogenic emissions as indicated in **Figure 2a**. Luanda is the capital of Angola with dense population, and the median surface NO$_2$ concentrations observed at this station ranged from 10 ppb to

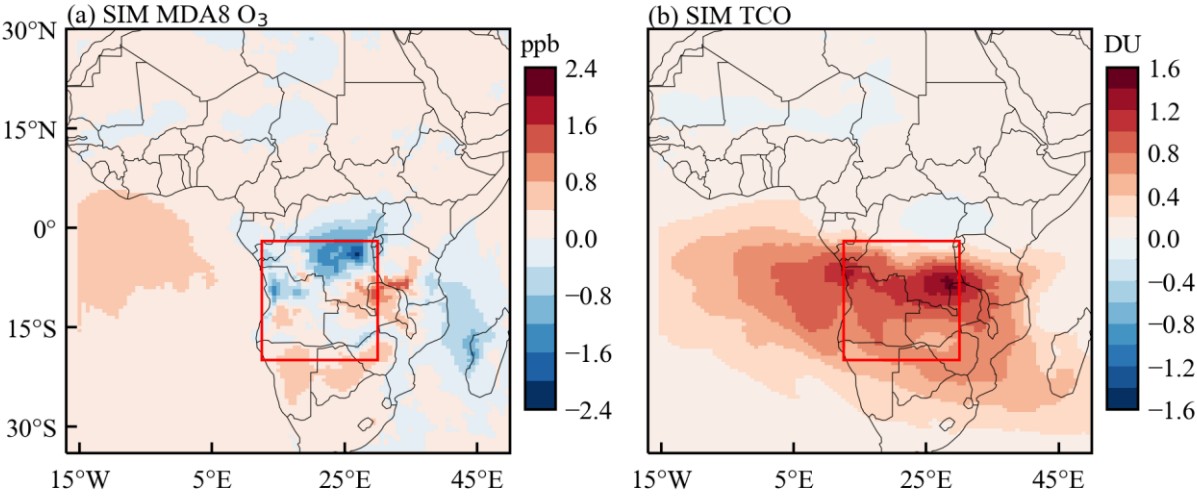

**Figure 8.** Effects of vertical partitioning of model BB emissions in surface MDA8 ozone and tropospheric ozone columns. The baseline simulation follows the vertical distribution of QFED2 emission (i.e., 65% emissions below the PBL and 35% emissions into the free atmosphere), and the sensitivity simulation allows 100% BB emissions emitted below the PBL. Here the plots are the differences between the baseline simulation (Run_QFED) and sensitivity simulation (Run_QFED_PBL).

**Table 3.** Monthly anthropogenic $NO_x$ emissions in Southern Africa and Luanda (unit: Gg month$^{-1}$).

|  |  | Jan. | Feb. | Mar. | Apr. | May | Jun. | Jul. | Aug. | Sep. | Oct. | Nov. | Dec. |
|---|---|---|---|---|---|---|---|---|---|---|---|---|---|
| Southern Africa | CEDSv2 | 18.2 | 16.5 | 18.2 | 17.6 | 18.2 | 17.6 | 18.2 | 18.2 | 17.6 | 18.2 | 17.6 | 18.2 |
|  | HTAPv3 | 27.1 | 24.9 | 17.6 | 17.0 | 17.3 | 18.9 | 19.0 | 17.1 | 16.7 | 17.8 | 17.0 | 16.4 |
| Luanda | CEDSv2 | 0.8 | 0.7 | 0.8 | 0.8 | 0.8 | 0.8 | 0.8 | 0.8 | 0.8 | 0.8 | 0.8 | 0.8 |
|  | HTAPv3 | 6.3 | 5.2 | 1.5 | 1.0 | 1.2 | 1.1 | 1.1 | 1.2 | 1.0 | 1.3 | 1.2 | 1.0 |

30 ppb. The Humpata station is located at Universidade Privada de Angola, where the observed $NO_2$ concentrations ranged from 5 ppb to 25 ppb, with large day-to-day variations of up to 20 ppb. The Luena station is located in a residential area of Luena, where the observed $NO_2$ concentrations were much higher than those of the previous two stations, with a maximum of 50 ppb. **Figure 9** shows the comparison of the observed and simulated daily surface $NO_2$ concentrations in Luanda, Humpata, and Luena, respectively. Compared with the observed values, the modeled $NO_2$ concentrations for all three cities are much lower than the observed values, underestimated by 90%. There is also a large underestimation of surface $NO_2$ in Luanda compared to the observations from Campos et al. (2021). This indicates that urban $NO_x$ emissions in our model are highly underestimated in Southern Africa, although the lack of model resolution accuracy is also a reason for the underestimation at the station scale.

To test the sensitivity of simulated $NO_2$ concentration to urban emissions, we increased the $NO_x$ emissions in the CEDSv2 inventory by a factor of 10, and the model results are shown in **Figure 9**. In Luanda, the Normalized Mean Bias (NMB) of simulated $NO_2$ concentrations will be decreased from -94% in the baseline simulation to -55%, while the changes of $NO_2$ in Humpata and Luena are very limited with an improvement of the NMB of only 5%. Even if when $NO_x$ emission is scaled up by a factor of 20, the simulated $NO_2$ concentrations in Humpata and Luena are increased by only 11-21%. For Luanda, there is a NMB of -50% in modeled surface $NO_2$ with 10-fold $NO_x$ emissions. But for Humpata and Luena, the CEDSv2 inventory is not capable to correctly estimate the anthropogenic sources, leading to the small sensitivity of simulated $NO_2$ concentration to perturbed urban emissions.

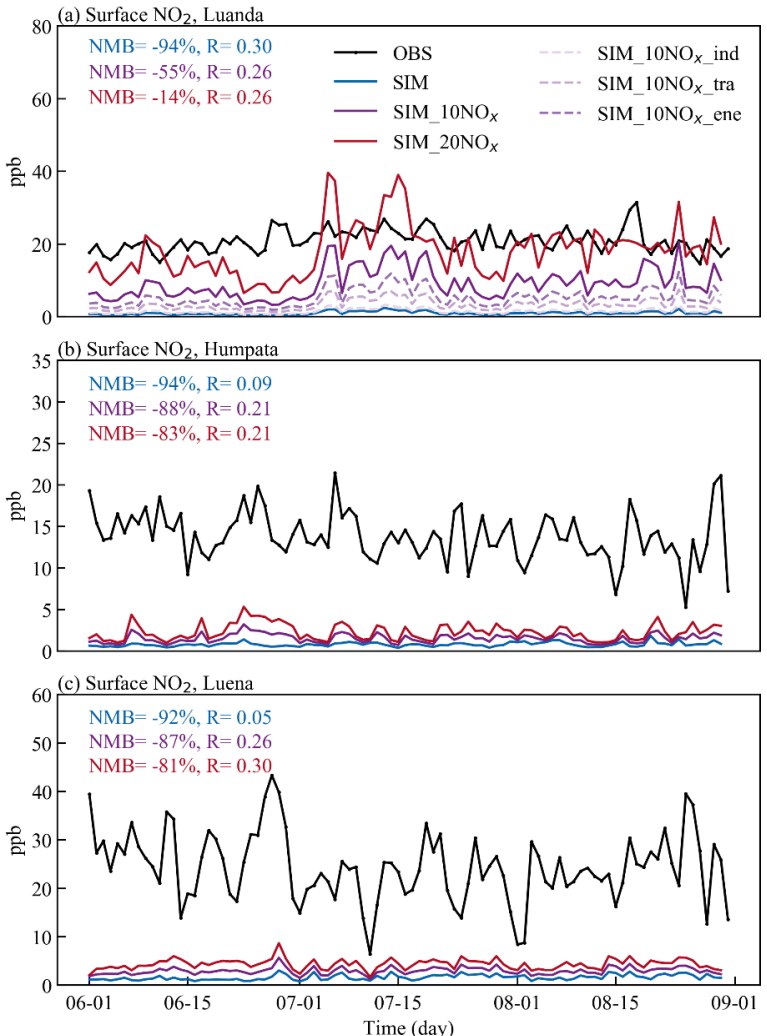

**Figure 9.** Time series of simulated and observed daily median surface $NO_2$ concentrations in Southern African cities (Luanda, Humpata, Luena) in June-August 2023. The model was driven by the QFED2 inventory and fixed CEDSv2 inventory in 2019. The "SIM" denotes the baseline simulation (Run_QFED_2023), and "SIM_10$NO_x$", and "SIM_20$NO_x$" denote the 10-fold and 20-fold increase in $NO_x$ emissions from CEDSv2. The dashed lines indicate a 10-fold increase in $NO_x$ emissions from the energy (ene), industry (ind) and transportation (tra) sectors, respectively, in the CEDSv2 inventory.

Although this study focused on ozone simulation, the comparison of model results against the valuable $PM_{2.5}$ measurements will be also meaningful to understand urban emissions in this region. **Figure 10** shows the time series of observed and simulated $PM_{2.5}$ concentrations in June-August 2023. The $PM_{2.5}$ concentrations observed at both the Humpata and Luanda

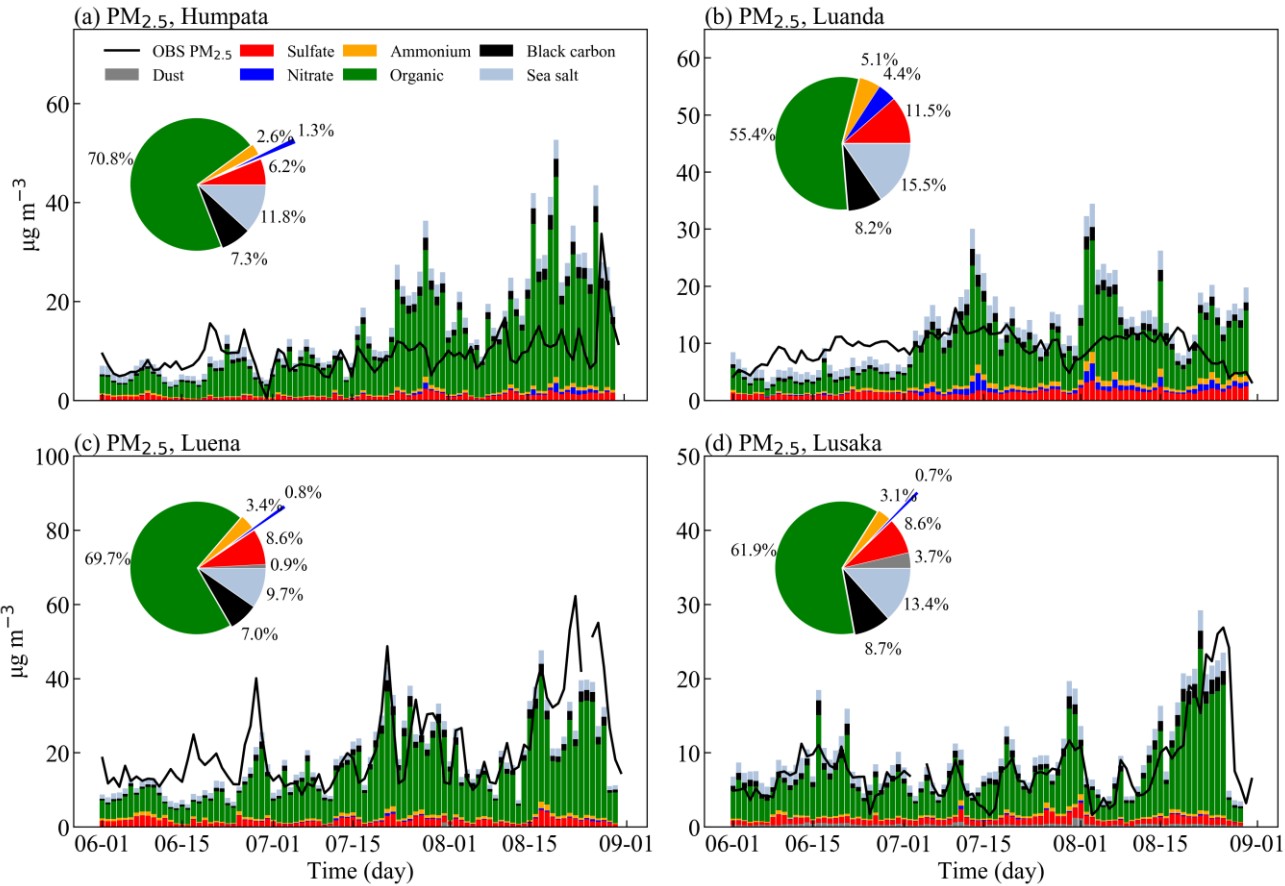

**Figure 10.** Comparison of simulated time series of PM$_{2.5}$ and its components against with the observed PM$_{2.5}$ (black line) during June-August 2023. (a) and (b) are for Humpata and Luanda, respectively, where we removed dust concentrations from the simulated PM$_{2.5}$ due to the large uncertainties in the model, and (c) and (d) are for Luena and Lusaka, respectively. The pie charts show the percentage contributions of each component to total PM$_{2.5}$ concentrations.

sites were around 10 μg m$^{-3}$. The PM$_{2.5}$ concentrations at the Luena site were slightly higher compared to the other two sites, with median concentrations ranging from 10 μg m$^{-3}$ to 70 μg m$^{-3}$. Lusaka is the capital of Zambia and the observed site is located within the urban area of Lusaka, where PM$_{2.5}$ concentrations were about 10 μg m$^{-3}$ in June-July and then suddenly increased to about 20 μg m$^{-3}$ in August. Figure S11 shows the comparison of simulated and observed PM$_{2.5}$ concentrations, and the model can capture the day-to-day variation in PM$_{2.5}$ concentrations at Luena as well as Lusaka sites, with NMBs of -12% and 24% and correlation coefficients of 0.7 and 0.87, respectively. But in Luanda and Humpata, there is a large overestimation in simulated PM$_{2.5}$ concentration and a large proportion of PM$_{2.5}$ components is contributed by dust, possibly due to the influence of the Namib and Kalahari Deserts (Nyasulu et al., 2023). We excluded dust

concentration in the calculation of total $PM_{2.5}$ concentration for the time being, due to its large uncertainties in the GEOS-Chem simulation (Weagle et al., 2018). After removing dust concentration, the NMB in the model will be reduced from 149% to 37% in Luanda.

In terms of $PM_{2.5}$ components, the highest contribution of OC to $PM_{2.5}$ concentrations is found at all the four sites, which can be attributed to the effects from biomass burning (Nyasulu et al., 2023). Secondary inorganic aerosols account for about 20% at Luanda and about 10% at other sites. In addition, we also compared the changes in $PM_{2.5}$ concentrations at each site after scaling up anthropogenic $NO_x$ emissions by a factor of 10, and found that $PM_{2.5}$ at the Luanda site can increase by up to 50 μg m$^{-3}$. In previous studies, changes in $PM_{2.5}$ concentrations in Southern Africa have often been attributed to BB (Nyasulu et al., 2023; Booyens et al., 2019). However, this study shows that anthropogenic emissions in Luanda could also have a great impact on $PM_{2.5}$ concentrations, highlighting the underappreciated role of anthropogenic emissions in urban air quality over the Southern Africa.

We further explored the sensitivity of ozone concentration to perturbed $NO_x$ emissions. Figure S12 shows the response of ozone concentration at each site after anthropogenic $NO_x$ was increased by 10 times. Relative to the baseline run, the ten-fold $NO_x$ simulation can increase ozone concentrations by 0-10 ppb in the Humpata and Luean regions. However, in Luanda there was increased ozone in June but decreased ozone in July-August in response to ten-fold $NO_x$ emissions, indicating that ozone chemistry in Luanda may be likely shift into transition regime with increasing emissions. Again, these results confirm that the underestimation of anthropogenic emissions in the urban areas of Southern Africa now can have an important impact on local ozone assessment.

Although there is a lack of surface ozone observations in Southern Africa that can be directly compared with our model results, we can conclude from the model evaluation against surface $NO_2$ and $PM_{2.5}$ measurements: 1) the large underestimation in modeled urban scale $NO_2$ in Southern Africa is mainly due to large low biases of $NO_x$ emission in the CEDSv2 inventory, i.e., a strong underestimation in Luanda and the misrepresentation of anthropogenic emission estimates in Humpata and Luena, 2) the model is able to capture the observed variations in $PM_{2.5}$ concentrations in the areas that are less affected by dust, and 3) the bias in anthropogenic emission inventories can strongly affect the assessment of $PM_{2.5}$ and ozone concentrations in urban Southern Africa.

### 3.3.3 Model evaluation against satellite measurements

Satellite observations were further used to support the deduction of the underestimated urban emissions. As the capital of Angola, Luanda is of much higher anthropogenic emissions compared to other cities in Southern Africa. In the following,

we focused on Luanda where satellite signals could be stronger to detect $NO_x$ emissions. **Figure 11** shows the simulated and satellite-based $NO_2$ columns for fire season (July-August 2019) and non-fire season (January-February 2020), respectively. To minimize the effects from background levels, here the $NO_2$ values are the columns at Luanda minus the mean columns averaged over the sea downwind. For July-August 2019, the urban $NO_2$ enhancement in Luanda simulated by the model was 26% lower than that observed by the TROPOMI; for January-February 2020, the simulated $NO_2$ enhancement was underestimated by 61% compared to TROPOMI. Due to the decreased contribution from anthropogenic sources to $NO_2$ columns during the fire season, the moderate underestimation during fire season (July-August 2019) in Luanda may be due to the long-term transport of pollutants from biomass burning to urban areas. As suggested by TROPOMI satellite, therefore, $NO_x$ emissions from CEDSv2 were underestimated by at least a factor of 2 in urban Luanda. At the same time, we find that the $NO_2$ enhancement in Luanda observed by OMI was 70% lower compared to TROPOMI and the OMI instrument cannot detect the high emissions in Luanda, demonstrating the advantage of TROPOMI instrument in observing regions with significant $NO_x$ spatial heterogeneity.

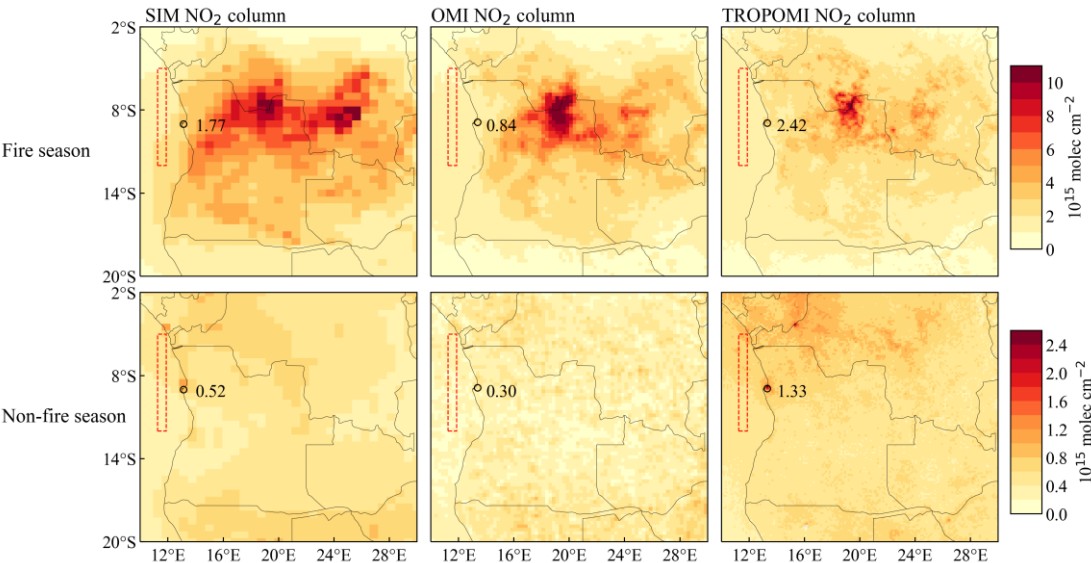

**Figure 11**. Spatial distribution of $NO_2$ columns from the model, OMI, and TROPOMI during the fire season (top, July-August 2019) and non-fire season (bottom, January-February 2020). Circles indicate the Luanda city and numbers around them indicate $NO_2$ column enhancement in the Luanda city. The dashed boxes indicate the downwind ocean region whose concentrations were subtracted to obtain the $NO_2$ column enhancement in Luanda.

To further identify the key emission sectors in Luanda, we perturbed the $NO_x$ emissions from three sectors (transportation, industrial, and energy) by a factor of 10 in **Figure 9a**. Surface $NO_2$ concentrations in Luanda responded better to changes in $NO_x$ emissions from the energy and transportation sectors, with NMB reduced by 20% and 11%, respectively. Figure S13 shows the $NO_2$ column changes in response to the emission perturbations. When all sources of $NO_x$ emissions in the CEDSv2 inventory were increased by a factor of 10, the simulated $NO_2$ column enhancement in Luanda will be 3-4 times the TROPOMI measurement. Namely, the ten-fold increase in $NO_x$ emissions to be consistent with surface measurement cannot reconcile with satellite measurements. In addition, the response of $NO_2$ column in Luanda to sectoral perturbations in $NO_x$ emissions is mainly linear (Figure S13). These model-satellite comparisons suggest an underestimation of $NO_x$ emissions from CEDSv2 by at least a factor of 2 in urban Luanda.

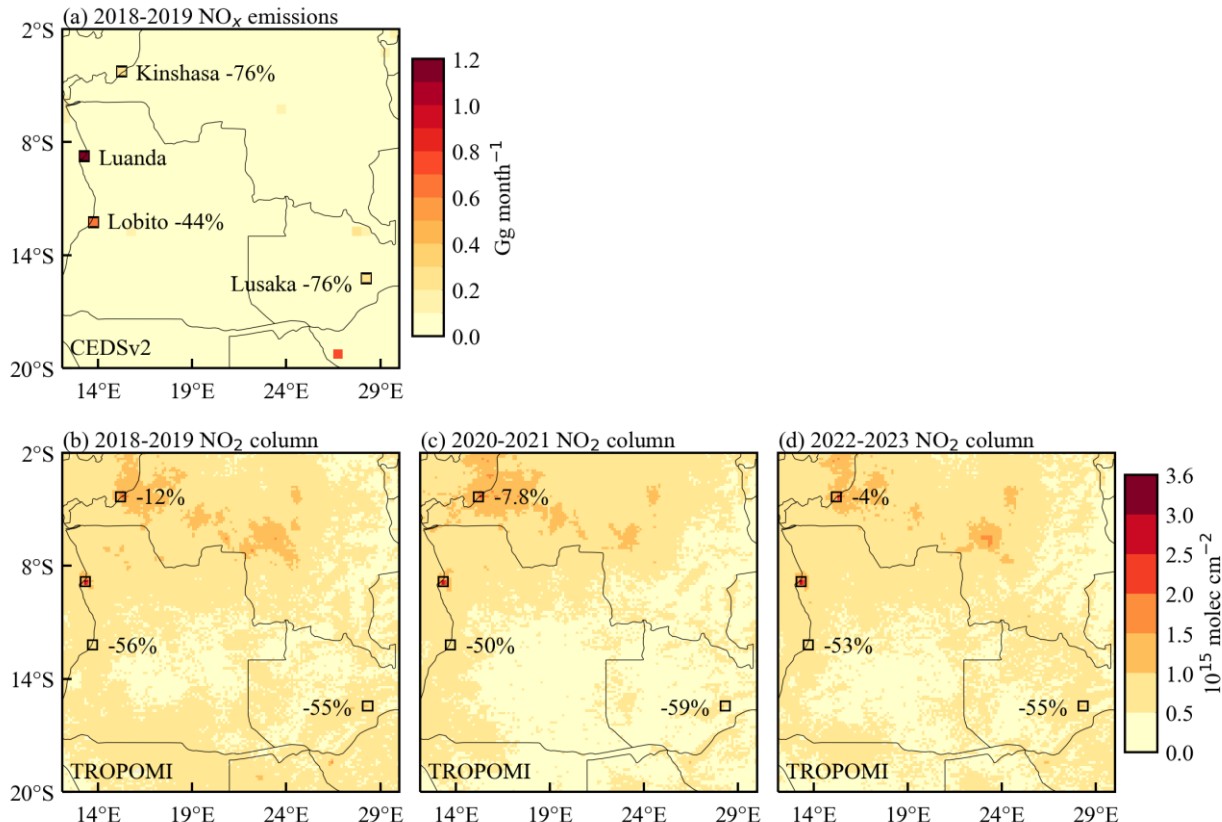

**Figure 12.** Anthropogenic $NO_x$ emissions from CEDSv2 and $NO_2$ columns from TROPOMI in typical cities in Southern Africa. (a) Spatial distribution of $NO_x$ emissions from the CEDSv2 inventory in March-April of 2018-2019. (b-d) Spatial distribution of $NO_2$ columns observed by TROPOMI in March-April, 2018-2023. All of the numbers in the plots are the percentage changes by taking Luanda as a reference.

To further explore the underestimation of anthropogenic emissions in Southern Africa, we use Luanda as a baseline to show the difference between anthropogenic $NO_x$ emissions and satellite $NO_2$ columns for other cities in **Figure 12**. The selected cities are grid cells with high $NO_x$ emissions in the CEDSv2 inventory. The tropospheric $NO_2$ during non-fire season is dominantly contributed by the lower atmosphere (Figure S14), and then we selected March-April 2018-2023 for comparison in order to exclude the effect of biomass burning. In Kinshasa (the capital of the DRC), anthropogenic $NO_x$ emissions are 76% lower than those in Luanda, but their difference in satellite $NO_2$ columns was only 4-12%, suggesting that anthropogenic emissions from Kinshasa were also underestimated; In Lusaka (the capital of Zambia), its $NO_2$ columns were 55 % lower than those in Luanda while the difference is 76% in anthropogenic $NO_x$ emissions. Combining satellite data with CEDSv2 $NO_2$ emissions provides additional evidence of the prevalent underestimation in anthropogenic $NO_x$ emissions in major cities over Southern Africa.

## 4. Conclusions and discussion

In this study, we focused on Southern Africa where tropospheric ozone levels were thought extremely high but have been less studied. By integrating the nested GEOS-Chem model and the newly-available surface and satellite observations to evaluate the tropospheric ozone levels and their main drivers in Southern Africa. In particular, we quantified the impact of biomass burning (BB) on tropospheric ozone at the regional scale in Southern Africa and the effects of anthropogenic emissions in urban ozone levels. This study provides a better understanding of the impacts of key emission sources on air quality modeling in Southern Africa, which will be also important for health risk assessment, climate change prediction, and sustainable strategy development.

The anomalously high values of dry-season tropospheric ozone in Southern Africa are mainly caused by BB, but there is a large discrepancy of a factor of 2-3 in estimated BB emissions among different inventories. Comparison of model simulations against satellite $NO_2$ observations revealed that the widely-used GFED4.1 inventory tends to strongly overestimate $NO_x$ emissions in Southern Africa, while model results with the QFED2 inventory were more consistent with observations. This is consistent with the finding by Anderson et al. (2021) that their model driven by the GFED4.1 inventory tended to overestimate $NO_2$ concentrations in the Africa, with a bias of about 100%. Consequently, the simulated regional surface MDA8 ozone was decreased from 74 ppb by GFED4.1 inventory to 56 ppb by QFED2 inventory, and accordingly the model bias in TCO was reduced from 14% to 2.3%. The modeled HCHO and CO columns were generally consistent between GFED4.1 and QFED2 inventories. Using the QFED2 inventory, we explored the impact of BB emission heights on ozone simulations and found that the effect of the vertical emission distribution was in the range of $\pm 2.4$ ppb for surface MDA8 ozone and from -0.4 to 1.6 DU for TCO over Southern Africa; in contrast, the difference in BB aerosol emissions between the inventories could affect ozone simulation strongly through aerosol chemistry.

We conducted further sensitivity experiments using the QFED2 inventory to explore the contribution of anthropogenic emissions. Compared with surface $NO_2$ and $PM_{2.5}$ observations, we found that the CEDSv2 anthropogenic inventory likely strongly underestimated anthropogenic emissions in typical Southern African cities and even incorrectly represented anthropogenic sources in some areas. Our study also found that the TROPOMI instrument performs effectively in these low-emission areas where there is a lack of observational data, while the OMI is unable to capture urban-scale hotspots of $NO_2$ columns over Southern Africa. We also demonstrated that urban ozone and $PM_{2.5}$ concentrations are strongly influenced by the underestimated anthropogenic emissions. For example, a ten-fold increase in anthropogenic $NO_x$ emissions can change ozone concentrations by up to 10 ppb and increase $PM_{2.5}$ concentrations by up to 50 µg m$^{-3}$ in some cities.

Although several studies examined high ozone levels in Southern Africa (Meyer-Arnek et al., 2005; V. Clarmann et al., 2007), they only highlighted the role of biomass burning but overlooked the role of anthropogenic emissions. Recent findings by Wiedinmyer et al. (2023) pointed out the large uncertainties in bottom-up BB emissions, but they failed to constrain the uncertainties due to the lack of observational data. Here we found that the difference among BB inventories can have a great impact on urban air quality assessment. In addition, with combined surface observations, satellite data and model simulations, we demonstrated for the first time that anthropogenic emission inventories are strongly low-biased in urban Southern Africa. It suggests the important impacts of anthropogenic emissions in Africa with increasing urbanization.

The rapid change in anthropogenic emissions is already affecting air pollution and health risks in Southern Africa (Health Effects Institute, 2022), as well as might have impacts on regional climate change (Fotso-Nguemo et al., 2023). Assessing and predicting the impacts of different emission sources on air quality and human health rely heavily on model simulations. The performance of these model estimations is significantly influenced by the accuracy of emission inventories. For example, our finding of overestimated biomass burning emissions and underestimated anthropogenic emissions can strongly affect the ozone source attribution over Southern Africa due to the nonlinear ozone chemistry. As shown in Figure S15, in the dry season of 2019, regional surface MDA8 ozone over Southern Africa was contributed by 11 ppb, 8.0 ppb, and 1.5 ppb from BB emissions, natural emissions (mainly biogenic VOC), and anthropogenic emissions, respectively. However, when anthropogenic $NO_x$ emissions were increased by a factor of 10, estimated contributions from natural and anthropogenic emissions will be increased to 9.0 ppb and 3.3 ppb, respectively. In particular, the ozone source attribution spatially varies depending on the levels of anthropogenic $NO_x$ emissions (**Figure 13**). This suggests an ignored but critical role of anthropogenic emissions in ozone levels over Southern Africa.

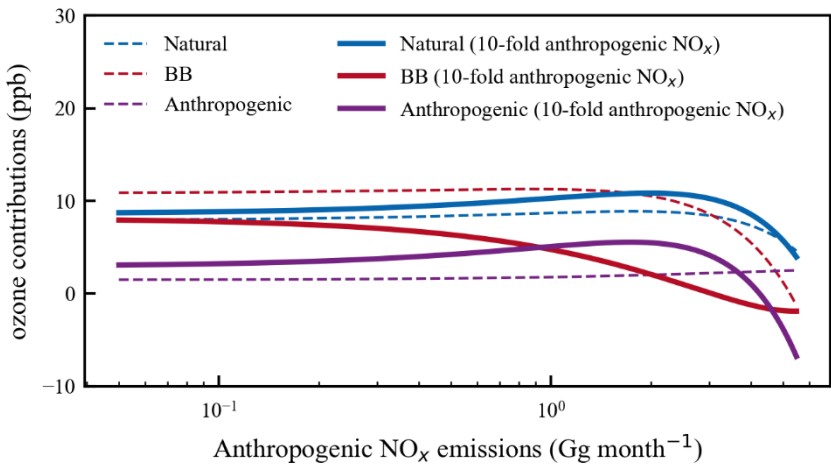

**Figure 13.** The simulated source contributions to surface ozone in July-August 2019 using the CEDSv2 emissions (dash lines) and 10-fold CEDSv2 NO$_x$ emissions (solid lines). Here the natural emissions refer to the biogenic VOC and soil NO$_x$ emissions. The x-axis is the anthropogenic NO$_x$ emissions in each grid cell and the y-axis the corresponding ozone contributions estimated from the sensitivity simulations.

There are also some uncertainties and limitations in our assessment of the major drivers of high ozone levels over Southern Africa. Firstly, due to the lack of observational data on surface ozone and VOCs, the effect of anthropogenic emissions on surface ozone over Southern Africa was explored by only comparing surface NO$_2$ concentrations, which may lead to biases in determining ozone chemical formation. Secondly, although we have used ozonesonde data from Ascension Island downwind of Southern Africa for comparison and the study by Jenkins et al. (2021) suggests that BB plumes in Southern Africa can have an impact on downwind regions, the long-range effects of BB emission on downwind urban regions were also not well validated due to the lack of vertical ozone measurement. Thirdly, the comparison between the model and satellite data could be improved by correcting the vertical profiles of chemical species in the process of satellite data. Fourthly, although the GEOS-Chem model has been shown to be able to capture spatial and temporal variations of ozone and its precursors over typical urban regions (Travis and Jacob, 2019), it is still challenging to capture the urban scale air quality in Southern Africa. Without accurate bottom-up anthropogenic inventories, we highlight the importance of high-resolution satellite observations for understanding air quality in developing regions such as Southern Africa.

Overall, this work provides a comprehensive understanding of the drivers and uncertainties of tropospheric ozone in Southern Africa, particularly the overestimation in BB emissions and the underestimation of anthropogenic emission inventories. More importantly, with more frequent BB and rising anthropogenic emissions in Africa, this study highlights the urgency of establishing the surface network for air quality measurement over Southern Africa. The more accurate

estimates of anthropogenic emission sources and more regular surface observations are the key to understand atmospheric chemistry over Southern Africa that is driven by rapidly changing anthropogenic and biomass burning emissions. The deepened understanding of major emission sources in Southern Africa will not only help us to provide a solid scientific basis for policymakers to effectively address air quality issues, but also will enhance the model capability to predict future air quality and climate change. In the future, anthropogenic air pollutants (e.g., $NO_x$ emissions) in Southern Africa under future scenarios are projected to increase all the way by 2060 (Figure S16); along with more fires under a warming future, Southern Africa will be a hotspot suffering from complex atmospheric chemistry and climate issues, presenting a grand challenge to realize the Sustainable Development Goals for having a healthy, climate-friendly, and resilient development in Africa. Our study serves as a baseline understanding of these key emission sources which are key drivers for modelling future air quality, climate change, and their socioeconomic impacts.

**Data availability**. Daily real-time air quality indexes for $NO_2$ and $PM_{2.5}$ were obtained from the Worldwide Air Quality Index (https://aqicn.org/station). The Ascension Island's ozonesonde data from the Southern Hemisphere Additional Ozone Sounding network (https://tropo.gsfc.nasa.gov/shadoz/). The OMI satellite data for $O_3$, $NO_2$ and HCHO are available at https://disc.gsfc.nasa.gov/datasets/. The TROPOMI data for $NO_2$ are available at https://www.earthdata.nasa.gov/sensors/tropomi. The MODIS data for AOD are available at https://ladsweb.modaps.eosdis.nasa.gov/search/. The MOPITT data for CO are available at https://giovanni.gsfc.nasa.gov/giovanni/.

**Author contributions.** KL designed the research and YW performed simulations and analyzed the results. XC, ZY, and MT helped the model simulations. PC, YY, XY, and HL contributed to the interpretation of the results. YW and KL wrote the paper with contributions from all co-authors.

**Competing interests.** The authors declare that they have no conflict of interest.

**Acknowledgements**. We appreciate the efforts of the China Ministry of Ecology and Environment with respect to supporting the national surface observation network and the publishing of hourly air pollutant concentrations. We also thank the Atmospheric Chemistry Modeling group at Harvard University for developing and managing the GEOS-Chem model, and the high-performance computing cluster Xuanwu maintained by the Atmospheric Chemistry & Climate Group at NUIST.

**Financial support**. This research has been supported by the National Natural Science Foundation of China (grant no. 42293323), the National Key Research and Development Program of China (2022YFE0136100), the Natural Science Foundation of Jiangsu Province (BK20240035), and Jiangsu Carbon Peak and Neutrality Science and Technology Innovation fund (BK20220031).

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
