# Peer review of "Revisiting the high tropospheric ozone levels over Southern Africa: Roles of biomass burning and anthropogenic emissions"

_EGUsphere, 2024_

## Author Comment (AC1)

**Response to Reviewer #1**

**General Comment:**

I enjoyed reading this paper. The results are relevant and interesting, and most of the figures are clearly labelled and easily interpretable. However, the authors should address a few points to improve the manuscript.

**Reply:** We greatly appreciate your constructive comments that have been carefully addressed and the point-by-point responses are in blue.

**Comments:**

Figure S1 and Table S1 should be moved to the main text. The reader needs to have an overview of the satellite instruments used in the study.

**Reply:** Thank you very much for your suggestion! Now we have moved Table S1 into the main text as Table 1 along with a description of the satellite data we used. As for Figure S1, it is the GEOS-Chem simulated surface ozone concentration from the benchmark experiment that can be publicly downloaded from https://ftp.as.harvard.edu/gcgrid/geoschem/1mo_benchmarks/. So we only described it in the introduction and did not move it into the main text.

**We added the description of the satellite data in Lines 142-149:** "The OMI sensor observes the globe once a day and is capable of obtaining the column concentration distributions of a variety of tropospheric trace gases (e.g., $NO_2$ and $O_3$). The TROPOMI sensor is a troposphere-specific observational instrument, and Wang et al. (2020) have compared the $NO_2$ concentrations of OMI and TROPOMI with observations. As for MODIS AOD, Shi et al. (2019) compared observations from 400 stations of the Aerosol Robotics Network (AERONET) with the MODIS AOD, and demonstrated that the MODIS was able to better capture the spatial and temporal variations of AERONET AOD (Zhang et al. 2024). MOPITT was launched in December 1999 on board the Earth observation satellite Terra with a 10:30 am equator crossing time (Kopacz et al., 2010)."

**Table 1:** Satellite and surface observations used in this study.

| Species | | Spatial resolution/ site locations | Observation period |
|---|---|---|---|
| $O_3$ | OMI | $1\,°\times1.25\,°$ | July-August 2019 |
| $NO_2$ | OMI | $0.25\,°\times0.25\,°$ | 2019-2020 |
| HCHO | OMI | $0.05\,°\times0.05\,°$ | July-August 2019 |
| $NO_2$ | TROPOMI | $0.125\,°\times0.125\,°$ | 2018-2023 |
| AOD | MODIS | $1\,°\times1\,°$ | July-August 2019 |
| CO | MOPITT | $1\,°\times1\,°$ | July 2019 |
| $PM_{2.5}$ | Humpata | (14 °34' S, 13 °26' E) | June-August 2023 |
| | Luanda | (8 °48' S, 13 °14' E) | |
| | Luena | (11 °45' S, 19 °54' E) | |
| | Lusaka | (15 °24' S, 28 °17' E) | |
| $NO_2$ | Humpata | (14 °34' S, 13 °26' E) | June-August 2023 |
| | Luanda | (8 °48' S, 13 °14' E) | |
| | Luena | (11 °45' S, 19 °54' E) | |
| $O_3$ | Ascension Island | (7 °58' S, 14 °24' W) | July-August 2017-2019 |

The results and discussion section should be reformulated to avoid starting the paragraphs/sentences with Figure X (for example, on lines 205, 212, …). The explanation of the figures should be integrated into the text.

**Reply:** We made some changes in the manuscript to avoid this issue.

Figure 11 should be changed to make the text in the figure more readable.

**Reply:** Thanks for the comment. We have removed numbers for the relative changes in Figure 11.

[Figure]

**Figure 11**. Spatial distribution of NO₂ columns from the model, OMI, and TROPOMI during the fire season (top, July-August 2019) and non-fire season (bottom, January-February 2020). Circles indicate the Luanda city and numbers around them indicate NO₂ column enhancement in the Luanda city. The dashed boxes indicate the downwind ocean region whose concentrations were subtracted to obtain the NO₂ column enhancement in Luanda.

**References:**

Kopacz, M., Jacob, D. J., Fisher, J. A., Logan, J. A., Zhang, L., Megretskaia, I. A., Yantosca, R. M., Singh, K., Henze, D. K., Burrows, J. P., Buchwitz, M., Khlystova, I., McMillan, W. W., Gille, J. C., Edwards, D. P., Eldering, A., V. Thouret, and Nedelec, P.: Global estimates of CO sources with high resolution by adjoint   inversion of multiple satellite datasets (MOPITT, AIRS, SCIAMACHY,TES), Atmospheric Chemistry and Physics, 10, 855–876, 2010.

Shi, H., Xiao, Z., Zhan, X., Ma, H., and Tian, X.: Evaluation of MODIS and two reanalysis aerosol optical depth products over AERONET sites, Atmospheric Research, 220, 75-80, 10.1016/j.atmosres.2019.01.009, 2019.

Wang, C., Wang, T., Wang, P., and Rakitin, V.: Comparison and Validation of TROPOMI and OMI NO2 Observations over China, Atmosphere, 11, 10.3390/atmos11060636, 2020.

Zhang, L., Wang, X., Huang, G., and Zhang, S.: Comprehensive Assessment and Analysis of the Current Global Aerosol Optical Depth Products, Remote Sensing, 16, 10.3390/rs16081425, 2024.

---

## Author Comment (AC2)

**Response to Reviewer #2**

In this study, the authors integrated the high-resolution GEOS-Chem model and newly-available measurements to estimate the impact of biomass burning (BB) and anthropogenic emissions on tropospheric ozone over Southern Africa. They identify the best estimate of BB emissions inventory and quantify the effect on regional tropospheric ozone over Southern Africa. The authors compare simulation outputs using different emission inventories. However, the discussion would be strengthened by providing a more in-depth coverage of the physical and chemical processes driving ozone and PM formation.

**Reply**: Thank you very much for your suggestion. We have carefully addressed the comments and the point-by-point responses are in blue.

**Comments:**

1. The authors should provide a summary of available surface observations and satellite data for chemical species and compare the model results with observations, including statistics on the spatial distribution.

**Reply:** Thank you very much for your suggestion.

We have added the surface observation data and the satellite data used in Table 1. Meanwhile, we compared the simulation results against satellite observations in Table S1, and the details of each surface site was described in the text.

**Table 1**. Satellite and surface observations used in this study.

| Species | | Spatial resolution/ site locations | Observation period |
|---------|---------|---------|---------|
| $O_3$ | OMI | $1°\times1.25°$ | July-August 2019 |
| $NO_2$ | OMI | $0.25°\times0.25°$ | 2019-2020 |
| HCHO | OMI | $0.05°\times0.05°$ | July-August 2019 |
| $NO_2$ | TROPOMI | $0.125°\times0.125°$ | 2018-2023 |
| AOD | MODIS | $1°\times1°$ | July-August 2019 |
| CO | MOPITT | $1°\times1°$ | July 2019 |

| | | | |
|---|---|---|---|
| PM$_{2.5}$ | Humpata | (14 °34' S, 13 °26' E) | June-August 2023 |
| | Luanda | (8 °48' S, 13 °14' E) | |
| | Luena | (11 °45' S, 19 °54' E) | |
| | Lusaka | (15 °24' S, 28 °17' E) | |
| NO$_2$ | Humpata | (14 °34' S, 13 °26' E) | June-August 2023 |
| | Luanda | (8 °48' S, 13 °14' E) | |
| | Luena | (11 °45' S, 19 °54' E) | |
| O$_3$ | Ascension Island | (7 °58' S, 14 °24' W) | July-August 2017-2019 |

**Table S1**. Statistics of spatial correlation coefficients between model simulation results and satellite data.

| | GFED4.1 | | QFED2 | |
|---|---|---|---|---|
| | NMB | R | NMB | R |
| OMI O$_3$ | -10.4% | 0.82 | -12.9% | 0.87 |
| OMI NO$_2$ | 22.0% | 0.83 | 9.3% | 0.92 |
| OMI HCHO | -2.9% | 0.79 | -5% | 0.76 |
| TROPOMI NO$_2$ | 8% | 0.78 | -3% | 0.91 |
| MODIS AOD | -34% | 0.9 | 5.7% | 0.89 |
| MOPITT CO | -17.4% | 0.89 | -17.1% | 0.89 |

2. In addition to the overall emission rate difference, how do spatial variations compare across the different emission inventories? A summary of statistics analysis would be helpful.

**Reply:** Thanks for this very helpful suggestion. We have compared their spatial differences in Figure S1 and some description in **Lines 293-300**:

"Spatially, there are also evident differences among different biomass burning inventories (**Figure S4**). The spatial distribution of the high values in GFED4.1 and QFED2 is generally consistent with a spatial correlation coefficient of 0.76, both showing high emissions in northeastern Angola. In contrast, the GFED5 inventory has high NO$_x$ emissions concentrated in southwestern Congo, and its spatial

distribution differs considerably with QFED2. The GFAS inventory has a similar spatial distribution with QFED2 (a correlation coefficient of 0.84), but GFAS cannot capture the localized high emissions as shown in QFED2 and GFED4.1. However, the FINNv1.5 and FINNv2.5 exhibit a very different spatial distribution compared to other inventories, with low emissions in Angola and high emissions in the Congo region. Their spatial correlation coefficients with the QFED2 inventory are 0.06 and 0.31, respectively."

[Figure]

**Figure S4.** Spatial distribution of monthly $NO_x$ emissions from different biomass burning emission inventories in July-August 2014.

3. BB not only emits NOx, but also VOCs and PM. The authors should summarize the related information such as CO, VOC, NOx, BC and OC, which were stated to play a role in ozone concentration. Are there any specific ratios among emitted chemical species?

**Reply:** We calculated the total CO, VOC, $NO_x$, BC and, OC emissions from the six BB inventories in **Figure S5** and have added them in **Lines 302-305**: "In addition to $NO_x$ emissions, the VOC emissions are the highest in GFED5 and FINNv2.5 inventories, and the other four inventories show much smaller VOC emissions. Each inventory adopts different specific ratios for emitted chemical species, but they also differ with each other. For example, there is a $NO_x$/OC ratio of 1:0.6 in GFED4.1, 1:1.5 in GFED5, GFAS, and FINNv1.5, 1:3 in QFED2, and 1:1 in FINNv2.5 (Figure S5)."

[Figure]

**Figure S5.** Estimated species emissions (Tg month$^{-1}$) in different biomass burning emission inventories in Southern Africa, July-August 2014.

4. Line 174: should "Run_QFED_34%" be corrected to "Run_QFED_66%NOx"?

**Reply:** Corrected.

5. Lines 227-228: When comparing ozone concentrations with Dewitt et al. (2019), the authors should present results for both GFED and QFED emissions at the grid point associated with the station location. Currently, only Run_GFED is presented.

**Reply:** We have added this result in **Lines 254-255:** "Based on our simulation results, it can be found that the daily maximum ozone during the BB season is 86 ppb for Rwanda in the Run_GFED run, compared to only 62 ppb in Run_QFED run."

6. Lines 235-240 and Figures 5(a)-(c): OMI O3 shows significantly higher ozone concentrations over the Atlantic Ocean compared to the simulation. Could this discrepancy be related to the meteorological conditions in the model? This issue might also influence the comparison of NOx concentrations between the model and observations in the studied cases.

**Reply:** This discrepancy is not caused by the MERRA-2 reanalysis meteorology in the model. And we

show that the higher OMI ozone concentrations over the Atlantic Ocean is mainly due to the different background ozone levels between the satellite and model simulation.

We have discussed this in **Lines 259-260**: "Also, in Figure S2, we find that the GEOS-Chem simulated (Run_QFED) and OMI tropospheric ozone columns are in good agreement over the Atlantic Ocean after individually subtracting the background ozone values."

[Figure]

**Figure S2.** The comparison of GEOS-Chem simulated (left and middle panels) and satellite-based (right panel) tropospheric ozone columns after individually subtracting the background ozone values averaged over the black box (-34~-25°S, 17°W~0).

7. Line 303: if the case with QFED2 NOx emissions reduced by 34% (Figure S3) better aligns with satellite TCO data, would the FINNv1.5 emission inventory, which has ~ 0.67 of QFFD2 NOx emission (Figure 4c), be a more appropriate NOx inventory for this study?

**Reply:** No, the FINNv1.5 inventory wouldn't be more appropriate due to its strong spatial biases. Although the FINNv1.5 $NO_x$ emissions are similar to the 67% of QFED2 $NO_x$ emissions, in **Figure S3** it can be found that the spatial distribution of the FINNv1.5 $NO_x$ emissions is quite different from other inventories, e.g., with a spatial correlation of 0.06 with the QFED2 inventory and of 0.29 with GFED4.1 inventory. After evaluating with the satellite $NO_2$, we demonstrate that the QFED2 inventory has a better regional representativeness of $NO_x$ emissions, as shown in **Table S1**.

8. Figures (g)-(l): the authors should address why the model predicts relatively low HCHO and CO concentrations.

**Reply:** We have explained this in the revision as follows:

**Lines 362-364:** "and the underestimated HCHO columns in GEOS-Chem might be due to some missing VOC species (Zhao et al., 2024) and the lower anthropogenic $NO_x$ emissions in Southern Africa that both affect the chemical production of HCHO."

**Lines 375-377:** "The regional average of CO column concentrations simulated by GEOS-Chem is underestimated by approximately 10% compared to MOPITT, which reflects a long-lasting issue of CO underestimation in GEOS-Chem model (David et al., 2019; Ni et al., 2018)."

9. Since pollutant concentrations can exhibit strong diurnal variation, was the simulation data aligned with the satellite overpass times in the region for the model-observation comparison?

**Reply:** Yes! We did sample the simulation results with the satellite overpass times, and have added this point in **Lines 152-153**: "We sampled the model simulation results consistent with satellite overpass times in the following comparisons."

10. Lines 318-319: how do BB VOC emissions in both emission inventories compare with anthropogenic (AVOC) and biogenic (BVOC) VOCs in Figure 3?

**Reply:** We have included BB VOC emissions in Figure 3, and added some comparison in **Lines 234-236**: "BB VOC has similar seasonal variability in both inventories, but the GFED4.1 inventory emits 2-3 times as much as the QFED2 inventory in fire season. The BVOC emissions are generally higher than BB VOC emissions except for those in July-August months from the GFED4.1 inventory."

[Figure]

**Figure 3.** Seasonal variations in anthropogenic $NO_x$ (deep blue), soil $NO_x$ (grey), biomass burning

NO$_x$ (red), biomass burning VOC (red), anthropogenic VOC (blue), and biogenic VOC (yellow) emissions in 2019 (unit: Gg month$^{-1}$). Anthropogenic NO$_x$ and VOC are from CEDSv2 inventory, soil NO$_x$ and BVOC are calculated by the GEOS-Chem model, and biomass burning NO$_x$ and VOC are from GFED4.1 and QFED2 inventory.

11. Lines 327: What are the major chemical species in BB VOCs, and how do they influence ozone formation beyond HCHO formation?

**Reply:** According to **Figure R1**, the major chemical species from BB VOC are OVOC, followed by alkanes, and alkenes; and OVOC and alkenes dominate the chemical formation of ozone and HCHO. However, we find the impact of uncertainties in BB VOC emission in ozone formation is much smaller than that from BB NO$_x$ emissions (Figure S6).

We have added this in **Lines 352-355**: "In contrast, we find that the BB VOC emissions from GFED4.1 inventory are about 3 times the QFED2 inventory in fire season, but the regional mean changes are only 2.5 ppb for MDA8 ozone and 0.94 DU for TCO for July-August 2019 in response to a tripled QFED2 VOC emissions (Figure S6)".

[Figure]

**Figure R1.** NMVOC components of the QFED2 inventory and their percentage contribution to formaldehyde and ozone.

[Figure]

**Figure S6**. Changes in surface MDA8 ozone (left) and TCO (right) when VOC emissions from QFED2 inventories are tripled for July-August 2019. The regional mean changes are 2.5 ppb for MDA8 ozone and 0.94 DU for TCO.

12. Lines 340-341: The authors briefly mention the model results without adequate discussion. A more detailed explanation of how aerosol chemical processes influence surface ozone concentrations would be helpful to illustrate the causality.

**Reply:** We have added the following discussion in **Lines 394-397:** "Aerosol chemistry mainly influences ozone formation by altering photolysis and heterogeneous processes. On the one hand, aerosol can change the shortwave radiation reaching the ground through scattering and absorption, which in turn affects the photolysis rate. On the other hand, aerosol can update reactive radicals (e.g., $HO_2$, nitrogen radicals) that are critical for ozone formation."

13. Figure 9 and the associated discussion: The authors should evaluate the comparison between observations and simulations. Could the higher observed NOx concentrations at the observation site compared to the simulation be due to the emissions being concentrated in a small area, whereas the model averages emissions over a larger grid? This could explain the lower simulated concentrations.

**Reply:** Yes. The location of the four surface observation sites is shown in Figure R2. The stations are primarily located alongside streets. As you mentioned, the observed pollutant emissions are concentrated in smaller areas, whereas the model averages the emissions over a larger grid, which is one of the reasons for the underestimation of the modeled results compared to the observations,

We have added this point in **Lines 458-459**: "although the lack of model resolution accuracy is also a reason for the underestimation at the station scale."

[Figure]

**Figure R2.** Location of four surface observation sites (Google Earth).

**References:**

David, L. M., Ravishankara, A. R., Brewer, J. F., Sauvage, B., Thouret, V., Venkataramani, S., and Sinha, V.: Tropospheric ozone over the Indian subcontinent from 2000 to 2015: Data set and simulation using GEOS-Chem chemical transport model, Atmospheric Environment, 219, 10.1016/j.atmosenv.2019.117039, 2019.

Ni, R., Lin, J., Yan, Y., and Lin, W.: Foreign and domestic contributions to springtime ozone over China, Atmospheric Chemistry and Physics, 18, 11447-11469, 10.5194/acp-18-11447-2018, 2018.

Zhao, T., Mao, J., Ayazpour, Z., González Abad, G., Nowlan, C. R., and Zheng, Y.: Interannual variability of summertime formaldehyde (HCHO) vertical column density and its main drivers at northern high latitudes, Atmospheric Chemistry and Physics, 24, 6105-6121, 10.5194/acp-24-6105-2024, 2024.

---

## Author Comment (AC3)

**Response to Reviewer #3**

The study evaluates several available NOx emission inventories from biomass burning and anthropogenic activities using GEOS-Chem sensitivity simulations against a few ground-based measurements and multiple satellite observations in Southern Africa. While the manuscript is readable, it lacks depth in scientific analysis. The conclusions are based solely on sensitivity simulations with altered emissions, ignoring other factors that might affect surface ozone, NOx concentrations, and vertical column densities. The authors seem to imply that GEOS-Chem is flawless except for the input emission inventories, which is obviously untrue. Additionally, when evaluating anthropogenic NOx emissions, the uncertainties of QFED2 are not mentioned, which could undermine the entire analysis. Thus, the current analysis is unconvincing, even if some conclusions might be correct. A more comprehensive analysis is needed to draw more robust conclusions.

**Reply:** We thank the reviewer for the constructive comments and suggestions, which are very helpful for improving the clarity and reliability of the manuscript.

Firstly, we fully agree with the reviewer that it is hard to quantify the BB and anthropogenic emissions by only evaluating model simulations against satellite observations. Our aim is to understand the key drivers of tropospheric ozone over the understudied Southern Africa by combining model simulation and multiple measurements. To highlight this aim, we have changed the title of this manuscript to "Revisiting the high tropospheric ozone over Southern Africa: role of biomass burning and anthropogenic emissions".

Although there are still some existing limitations in the current version, we have greatly improved the manuscript by (1) comparing the role of BB $NO_x$ and VOC in ozone formation (Figure S6) and confirming the dominated contribution of BB $NO_x$ emissions to simulated ozone difference; 2) highlighting the spatial representativeness of different BB inventories supported by satellite measurements (Figure S4 and Table S1); 3) clarifying the seasonal and vertical dependence of simulated ozone to BB emissions (Figure S3 and Figure S7).

In short, relative to previous work by global models, our study provides a deeper understanding of tropospheric ozone formation from regional to urban scale over the Southern Africa by combining a set of nested chemical transport model simulations and the available measurements. Please find our detailed response (in blue) in the following.

Line 23: Please provide the full words of GFED4.1 at its first appearance, similar to other acronyms throughout the manuscript.

**Reply:** Added.

Line 34: "high-quality" to "high-resolution"

**Reply:** Changed.

Line 42-43: What do you mean by the photochemical oxidation of nitrogen oxides? Which species is NO2 oxidized to? Please change "oxidation" to "reactions."

**Reply:** We meant the photochemical reactions of $NO_x$, and we have changed it to "photochemical reactions" in **Line 43**.

Line 55: Delete "emissions" and change "emit" to "emits."

**Reply:** Changed.

Line 107: Delete the last "The."

**Reply:** Corrected.

Line 130: "there" to "these"?

**Reply:** Corrected.

Lines 141-144: It would be better to clarify the uncertainties of these satellite datasets.

**Reply:** Thank you for your suggestion. We have added some discussions on the uncertainties of satellite data in **Lines 153-156**: "Although these satellite datasets have been well employed to reflect emission changes, their uncertainties are also notable due to biases in slant column density, air mass factor, and stratosphere-troposphere separation. For example, the reported uncertainties in $NO_2$ columns from OMI and TROPOMI are 25-50% and they can be increased to 50-100% in terms of OMI HCHO columns.".

Lines 167-189: Why are these sensitivity simulations conducted in different years? Are you sure these

simulations are consistent, considering the natural variability of climate? In Line 169, you mentioned simulations from 2019-2023, but I didn't find any corresponding simulations in Table 1.

**Reply:** Considering the availability of observation data, we conducted these sensitivity simulations for different years with varying emissions and meteorology. But we didn't run the model consecutively from 2019 to 2013. To avoid misunderstanding, we have revised the manuscript with the following updates:

**Line 184**: "Here we focused our experiments on July-August 2019 for all sensitivity and benchmark simulations."

**Lines 186-188**: "Firstly, we used GFED4.1 and QFED2 inventories to simulate the hourly air pollutant concentrations for July-August, 2019 (Run_GFED and Run_QFED), respectively, and validated the model results with satellite observations."

**Line 195**: "Considering that surface air pollutant measurements are only available for June-August 2023, and then …"

Figure 2. Why didn't you use soil NOx and BVOC emissions from your simulations but those from Offline documents? Soil NOx and BVOC are sensitive to meteorological conditions and are calculated online in GEOS-Chem.

**Reply:** In the GEOS-Chem model, the offline soil $NO_x$ and BVOC emission inventories that were archived from previous model runs are the same with our online calculated emissions.

We have revised the caption of Figure 2: "Anthropogenic $NO_x$ and VOC are from CEDSv2 inventory, soil $NO_x$ and BVOC are calculated by the GEOS-Chem model, and biomass burning $NO_x$ and VOC are from GFED4.1 and QFED2 inventory."

Lines 234-235: Compared to what? The observed 70 ppb? Do the simulation results and observations match in timing?

**Reply:** The observation time for the Rwanda site is 2015-2017, and 70 ppb is the daily maximum ozone concentration at that observation time. Our simulation period is July-August 2019. We have noted this in **Line 255**: "Compared to the observed ozone in Rwanda, it may indicate…".

Lines 238-239: I don't understand the logic. How can the simple comparisons above support such a conclusion? How about the surface ozone seasonal variations? Are surface ozone concentrations lower

in the non-fire season?

**Reply:** Surface ozone in this region is characterized by strong seasonal variation with high values in the fire season and low values in the non-fire season, supported by model simulations and ozonesonde measurements: 1)We compared the simulated surface ozone concentrations during the fire season (July-August 2019) with those during the non-fire season (January-February 2020) in **Figure S3**. It shows the high ozone of 70 ppb during the fire season is over Southern Africa, while during the non-fire season ozone concentrations in Southern Africa are only around 35 ppb. 2) Based on the vertical distribution of seasonal ozone over Ascension Island, UK in 2018, the seasonal enhancement is evident with the highest summer ozone in the lower atmosphere (**Figure R1**).

To clarify this argument, we have added in **Lines 267-269**: "Considering the strong seasonal variation of surface ozone in Southern Africa (**Figure S3**) and the estimated ozone precursors from different sources in Figures 3, here the large differences in simulated surface ozone with different BB inventories demonstrate that BB contributes greatly to high ozone concentrations during the fire season in Southern Africa."

[Figure]

**Figure S3.** Spatial distribution of surface ozone during the fire season (July-August 2019) and non-fire season (January-February 2020) simulated by GEOS-Chem.

[Figure]

**Figure R1**. Vertical distribution of seasonal ozone over Ascension Island, UK, 2018.

Lines 244-245: I wonder how you processed the observed and modeling data. Aren't coincident observations and model results used in the comparison? Did you just calculate the seasonal or monthly mean model values regardless of the availability of observations?

**Reply:** We sampled the model data with the same satellite overpass times for comparison and excluded the simulation results with missing satellite measurement.

We noted this point in **Lines 152-153**: "We sampled the model simulation results consistent with satellite overpass times in the following comparisons."

Line 263: It can only explain the lower NOx emissions of FINNv1.5. How about GFAS?

**Reply:** We have added this discussion for GFAS in **Lines 288-290:** "This lower estimate in the bottom-up FINNv1.5 inventory may be attributed to the underestimated burned area and emissions (Wiedinmyer et al., 2011), and the lower top-down GFAS estimate could be due to a smaller emission factors (Liu et al., 2020)."

Line 271: "he" to "the".

**Reply:** Corrected.

Line 271-272: You can't make such a conclusion based on a single set of sensitivity tests with perturbed BB NOx emissions, although the conclusion may be correct.

**Reply:** We agree with the referee and have revised this argument to in **Lines 311-314**: "This remarkable discrepancy suggests that the uncertainties in BB emissions could play an important role in simulating surface ozone over Southern Africa. This is consistent with previous work that BB emissions lead to strong ozone increases in Southern Africa during the fire season (V. Clarmann et al., 2007; Jaffe and Wigder, 2012)."

Line 276: Delete "whether" and add "simulating" before "in."

**Reply:** Changed.

Line 282-283: The differences are also significant in the mid-troposphere. Did you calculate which part in altitude contributes most to the TCO difference between GFED4.1 and QFED2? It can't be directly derived from the vertical profiles in Figure6.

**Reply:** Thanks for the comment. Yes, as seen in Figure S6, the difference in tropospheric ozone concentrations between GFED4.1 and QFED2 is also notable at 3-6 km. This altitude is also the maximum contributor to TCO in this region. The ozone vertical profiles in Ascension Island simulated by the two BB inventories in Figure 6 also differ notably at this altitude range.

We have revised the text in **Lines 326-328**: "The differences in ozone vertical distribution due to the two BB inventories are notable in the troposphere below 6 km, in particular at the altitude range of 3-6 km (Figure S7). Compared to the ozonesonde observations, this bias can be also found while GEOS-Chem captures the vertical ozone variations well regardless of which inventory is used. This is consistent with the results of small TCO differences in Figures 5a-5b."

[Figure]

**Figure S7.** Vertical profiles of column concentrations across the latitude of -8°S simulated by GEOS-Chem using the GFED4.1 and QFED2 inventories, with the dashed line indicating the location of Ascension Island.

Line 296-305: NO2 vertical column retrieval is sensitive to the NO2 vertical profile. Did you redo the retrieval using your simulated NO2 vertical profiles?

**Reply:** Thanks for your suggestion. Due to the multiple satellite products, we didn't particularly recalculate the $NO_2$ columns using the GEOS-Chem simulated vertical profiles. We have pointed out this issue in **Lines 350-351**: "It is noted that this comparison between the simulated and satellite-based tropospheric columns could be biased due to their different representativeness in vertical profiles of chemical species."

Line 410-411: I'm afraid I must disagree with such a derivation.

**Reply:** We have removed this argument.

Line 454: "high" to "large low"

**Reply:** Changed.

Line 467-468: I don't understand the logic here. The sea downwind is also susceptible to BB's long-range transport.

**Reply:** Contribution from anthropogenic sources to $NO_2$ columns in Luanda should be reduced during

the fire season, so we attempted to attribute the smaller underestimation of NO₂ in Luanda during fire season to the long-term transport of BB emissions. To make it clearer, we have reorganized the text as follows:

**Lines 535-537**: "Due to the decreased contribution from anthropogenic sources to NO₂ columns during the fire season, the moderate underestimation during fire season (July-August 2019) in Luanda may be due to the long-term transport of pollutants from biomass burning to urban areas."

Line 472: "like Southern Africa" to "with significant NOx spatial heterogeneity."
**Reply:** Changed!

Lines 493-494: Did you check NO2 in the free troposphere? A large portion of NO2 lies in the free troposphere, significantly contributing to NO2 VCD.

**Reply:** Following your suggestion, we plotted the vertical NO₂ profiles simulated by GEOS-Chem for during fire season and non-fire season (**Figure S15**). During the non-fire season, NO₂ concentrations are only below 1.5 km; during the fire season, the important contribution to NO₂ columns can extend around 7 km. As such, for the analysis in this section of Figure 12, we selected March-April 2018-2023 for comparison in order to exclude the effect of biomass burning and to more accurately reflect the bias of anthropogenic emission inventories in the region.

We have added this point in **Lines 361-363**: "The tropospheric NO₂ during non-fire season is dominantly contributed by the lower atmosphere (Figure S14), and then we selected March-April 2018-2023 for comparison in order to exclude the effect of biomass burning."

[Figure]

**Figure S14.** Vertical profiles of $NO_2$ across the latitude of -9°S (Luanda is located at 8°80'S, 13°23'E) simulated by GEOS-Chem during the fire season (July-August 2019) and non-fire season (January-February 2020), with the dashed line indicating the location of Luanda.

**References:**

Jaffe, D. A. and Wigder, N. L.: Ozone production from wildfires: A critical review, Atmospheric Environment, 51, 1-10, 10.1016/j.atmosenv.2011.11.063, 2012.

Liu, T., L.J. Mickley, M.E. Marlier, R.S. DeFries, M.F. Khan, M.T. Latif, and A. Karambelas (2020). Diagnosing spatial biases and uncertainties in global fire emissions inventories: Indonesia as regional case study. Remote Sens. Environ., 237, 111557.

v. Clarmann, T., Glatthor, N., Koukouli, M. E., Stiller, G. P., Funke, B., Grabowski, U., Hopfner, M., Kellmann, S., Linden, A., Milz, M., Steck, T., and Fischer, H.: MIPAS measurements of upper tropospheric $C_2H_6$ and $O_3$ during the southern hemispheric biomass burning season in 2003, Atmospheric Chemistry and Physics, 7, 5861–5872, https://doi.org/10.5194/acp-7-5861-2007, 2007.

---

## Author Response (AR2)

**Response to Editor's comments**

The authors have revised the manuscript substantially and addressed the reviewers' concerns adequately. The manuscript in terms of scientific content shall be published as it is, but there are some minor typological and grammatical errors that should be fixed before the manuscript can be published. Specifically:

1. Please consider changing the title of the paper to: "Revisiting the high tropospheric ozone levels over Southern Africa: Roles of biomass burning and anthropogenic emissions".

Many thanks for the suggestion! We have revised the title of the manuscript as suggested.

2. There are various typological and grammatical errors in the newly added and modified texts, e.g., "due to a smaller emission factors", "Figures 3", "consistent with previous work that BB emissions...". Please conduct a thorough grammatical and formatting check on the newly added texts.

Thanks for checking! We have walked through the text to correct the typos.

**Response to Editorial comments from validation review**

1. For the next revision please include the section "Author`s contribution" which is mandatory. More information please see: https://www.atmospheric-chemistry-and-physics.net/submission.html#manuscriptcomposition 2. Coloured or marked text in *.pdf manuscript file is not allowed. Please provide a clean version of *pdf manuscript file (with black text).

We have added the section for Author's contribution and the text of manuscript file is now in black now.